# Re-expression of SMARCA4/BRG1 in small cell carcinoma of ovary, hypercalcemic type (SCCOHT) promotes an epithelial-like gene signature through an AP-1-dependent mechanism

Krystal Ann Orlando[1,2], Amber K Douglas[2], Aierken Abudu[3], Yemin Wang[4], Basile Tessier-Cloutier[4,5], Weiping Su[6], Alec Peters[6], Larry S Sherman[6,7], Rayvon Moore[2], Vinh Nguyen[2,8], Gian Luca Negri[9], Shane Colborne[9], Gregg B Morin[9,10], Friedrich Kommoss[11], Jessica D Lang[12], William PD Hendricks[12], Elizabeth A Raupach[12], Patrick Pirrotte[13], David G Huntsman[4,14], Jeffrey M Trent[12], Joel S Parker[2,15], Jesse R Raab[2,15]*, Bernard E Weissman[1,2]*

[1]Department of Pathology and Laboratory Medicine, University of North Carolina at Chapel Hill, Chapel Hill, United States; [2]Lineberger Comprehensive Cancer Center, University of North Carolina at Chapel Hill, Chapel Hill, United States; [3]Department of Microbiology and Molecular Genetics, Michigan State University, East Lansing, United States; [4]Department of Pathology and Laboratory Medicine, University of British Columbia and Department of Molecular Oncology, British Columbia Cancer Research Institute, Vancouver, Canada; [5]Department of Molecular Oncology, British Columbia Cancer Research Institute, Vancouver, Canada; [6]Division of Neuroscience, Oregon National Primate Research Center, Oregon Health & Science University, Beaverton, United States; [7]Department Cell, Developmental and Cancer Biology, Oregon Health & Science University, Portland, United States; [8]Curriculum in Toxicology and Environmental Medicine, University of North Carolina at Chapel Hill, Chapel Hill, United States; [9]Michael Smith Genome Science Centre, British Columbia Cancer Research Institute, Vancouver, Canada; [10]Department of Medical Genetics, University of British Columbia, Vancouver, Canada; [11]Synlab MVZ Pathologie, Mannheim, Germany; [12]Division of Integrated Cancer Genomics, Translational Genomics Research Institute (TGen), Phoenix, United States; [13]Collaborative Center for Translational Mass Spectrometry, Translational Genomics Research Institute (TGen), Phoenix, United States; [14]Department of Obstetrics and Gynaecology, University of British Columbia, Vancouver, Canada; [15]Department of Genetics, University of North Carolina at Chapel Hill, Chapel Hill, United States

*For correspondence:
jesse_raab@med.unc.edu (JRR);
weissman@med.unc.edu (BEW)

Competing interests: The authors declare that no competing interests exist.

**Abstract** Small cell carcinoma of the ovary, hypercalcemic type (SCCOHT) is a rare and aggressive form of ovarian cancer. SCCOHT tumors have inactivating mutations in *SMARCA4* (BRG1), one of the two mutually exclusive ATPases of the SWI/SNF chromatin remodeling complex. To address the role that BRG1 loss plays in SCCOHT tumorigenesis, we performed integrative multi-omic analyses in SCCOHT cell lines +/- BRG1 reexpression. BRG1 reexpression induced a gene and protein signature similar to an epithelial cell and gained chromatin accessibility sites correlated with other epithelial originating TCGA tumors. Gained chromatin accessibility and BRG1

recruited sites were strongly enriched for transcription-factor-binding motifs of AP-1 family members. Furthermore, AP-1 motifs were enriched at the promoters of highly upregulated epithelial genes. Using a dominant-negative AP-1 cell line, we found that both AP-1 DNA-binding activity and BRG1 reexpression are necessary for the gene and protein expression of epithelial genes. Our study demonstrates that BRG1 reexpression drives an epithelial-like gene and protein signature in SCCOHT cells that depends upon by AP-1 activity.

## Introduction

Small cell carcinoma of the ovary, hypercalcemic type (SCCOHT) is a rare and aggressive form of ovarian cancer affecting young women (mean age 24 years old) (*Young et al., 1994*; *Estel et al., 2011*). SCCOHT tumors are poorly differentiated with no known cell of origin (*Young et al., 1994*). However, over 94% of SCCOHT tumors have mutations and concomitant protein loss of *SMARCA4* (BRG1), one of the two mutually exclusive ATPases of the SWI/SNF chromatin remodeling complex (*Karnezis et al., 2016*; *Clarke et al., 2016*; *Ramos et al., 2014a*; *Ramos et al., 2014b*; *Kupryjańczyk et al., 2013*; *Witkowski et al., 2014*; *Jelinic et al., 2014*). Expression of *SMARCA2* (BRM), the other ATPase, is epigenetically silenced in SCCOHT, making the dual loss of both ATPases sensitive and specific for the diagnosis of SCCOHT (*Karnezis et al., 2016*). Unlike most other adult cancers with BRG1 mutations, such as non-small cell lung cancer (NSCLC), SCCOHT tumors have a simple genetic background and rarely have secondary mutations in other cancer driving genes (*Karnezis et al., 2016*; *Clarke et al., 2016*; *Ramos et al., 2014a*; *Ramos et al., 2014b*; *Kupryjańczyk et al., 2013*; *Witkowski et al., 2014*; *Jelinic et al., 2014*; *Lin et al., 2017*). Therefore, SCCOHT provides a unique model to understand the role that individual loss of BRG1 and the SWI/SNF complex plays in tumor development.

The SWI/SNF complex is involved in a variety of cellular processes, such as differentiation and cell cycle control and also interacts with transcription factors (*Kadam, 2000*; *Minderjahn et al., 2020*; *Lee et al., 2002*), such as CMYC (*Cheng et al., 1999*; *Weissmiller et al., 2019*) and AP-1 (*Vierbuchen et al., 2017*). AP-1 (activator protein-1) is a dimeric transcription factor involved in many cellular processes that does not display cell type specificity (*Sheffield et al., 2013*; *Eferl and Wagner, 2003*; *Garces de los Fayos Alonso et al., 2018*). AP-1 exists as either a homodimer or heterodimer transcription factor made up of a variety family members: FOS (c-FOS, FOSB, FRA1, FRA2), JUN (C-JUN, JUNB, JUND), ATF (ATF2, ATF3, ATF4, ATF5, ATF6, ATF6B, ATF7, BATF, BATF2, BATF3, JDP2), and MAF (c-MAF, MAFA, MAFB, MAFF, MAFG, MAFK) (*Eferl and Wagner, 2003*; *Garces de los Fayos Alonso et al., 2018*). All AP-1 family members share a characteristic basic leucine-zipper domain (bZIP) where the DNA-binding site is located (*Eferl and Wagner, 2003*; *Garces de los Fayos Alonso et al., 2018*).

While the first interaction described between AP-1 and SWI/SNF acted to repress transcription of c-FOS by BRG1 (*Murphy et al., 1999*), subsequent studies indicate positive interactions at enhancers and with more SWI/SNF subunits, such as ARID1A/B (*Mathur et al., 2017*; *Raab et al., 2015*; *Kelso et al., 2017*), SNF5 (*Alver et al., 2017*), BRM (*Gao et al., 2019*), and BRG1 (*Vierbuchen et al., 2017*; *Xue et al., 2019a*; *Hu et al., 2011*; *Michel et al., 2018*). Despite many noted interactions between AP-1 and BRG1 specifically, the biology driven by these two proteins in human cancer cells remains to be elucidated.

To better understand the role that BRG1 loss has in SCCOHT tumorigenesis, we reexpressed BRG1 in two SCCOHT cell lines (BIN67 and SCCOHT-1) and performed multi-omic analyses (ATAC-seq, RNA-seq, CUT and RUN, and proteome profiling). We found that BRG1 reexpression in BIN67 cells induced an epithelial-like gene and protein expression program along with a global gain in chromatin accessibility at distal regions. Sites of gained chromatin accessibility were highly enriched for transcription factor motifs from AP-1 family members and strongly recruited BRG1. Highly upregulated epithelial genes also contained AP-1 motifs at the transcription start site and recruited BRG1. Sites that contain AP-1 motifs and recruit BRG1 also recruited c-Jun in both cell lines. Using a dominant-negative AP-1 cell line, we found that the expression of AP-1-driven epithelial genes depended upon both AP-1 activity and BRG1 reexpression in BIN67, while neither alone was sufficient to induce protein expression. Although the current study supports previous reports of SWI/

SNF and AP-1 associations, it establishes a new biological connection between AP-1 and BRG1 to drive an epithelial cell-like differentiation in SCCOHT cell lines.

## Results

### BRG1 reexpression induces global morphological, transcriptomic, and epigenetic changes in BIN67 cells

In order to understand the global consequences of BRG1 loss in SCCOHT, we performed multi-omic analyses in parallel for the SCCOHT cell line BIN67 +/- BRG1 reexpression (*Figure 1a*). ATAC-seq and RNA-seq were performed from the same experimental samples, while CUT and RUN and proteomic analyses were performed from separate experimental groups (*Figure 1a*). In BIN67, the reexpression of BRG1 was comparable and consistent to endogenous protein expression levels in other cancer cell lines, such as HeLa, MCF-7, and 293FT, whereas other SCCOHT cell lines, SCCOHT-1, expressed moderate levels of BRG1 following reexpression (*Figure 1—figure supplement 1B, C*). Following BRG1 reexpression, BIN67 showed the previously reported elongated, spindle-like morphology (*Karnezis et al., 2016*; *Wang et al., 2017*; *Wang et al., 2018*) (white arrows, *Figure 1b*; *Figure 1—figure supplement 1A*), whereas cells transfected with control vector displayed a similar morphology to parental cells (*Figure 1b*; *Figure 1—figure supplement 1A*). Consistent with our previous findings, BRG1 induction also resulted in growth suppression (*Karnezis et al., 2016*), indicated by the reduced number of cells relative to the controls (*Figure 1b*; *Figure 1—figure supplement 1A*).

BRG1 has a well-established role in regulating chromatin remodeling and gene expression. Thus, we hypothesized that its reexpression would alter chromatin accessibility along with gene and protein expression. To answer this question, we preformed RNA-seq, ATAC-seq, and global proteome profiling. Unsupervised analysis of RNA-seq samples (principal component analysis [PCA]) showed that control samples segregated from the BRG1 reexpressed samples and clustered closely within respective treatment groups (*Figure 1—figure supplement 2A*). In the BIN67 RNA-seq data, 18,507 protein coding genes were identified, with an almost equal number of genes increasing (4087 genes, 22%) and decreasing (3444 genes, 19%) (significance cut off padj <0.05) (*Figure 1c*; *Supplementary file 1*). In comparison to the decreasing genes, genes that increased were at a greater log2 fold change and significance (*Figure 1c*). Interestingly, BRG1 (*SMARCA4*) was not the most abundantly expressed gene in the BRG1 reexpressed samples (BRG1/Control Log2FoldChange = 5.34), (*Figure 1c*; *Supplementary file 1A*). As expected, many genes previously classified as BRG1 target genes such as, *CD44*, *EHF*, *IFI16*, and *MMP7* increased in expression after BRG1/SMARCA4 reexpression (*Figure 1c*; *Supplementary file 1A*; *Strobeck et al., 2001*; *Reisman et al., 2002*; *Orvis et al., 2014*; *Strobeck et al., 2002*).

We next questioned whether similar changes in the proteome occur following BRG1 reexpression that were observed with the transcriptome. Unsupervised analysis of proteomic samples showed segregation of control and BRG1 samples (*Figure 1—figure supplement 2B*). Of the 5726 identified proteins, 306 proteins were significantly upregulated and 233 proteins were significantly downregulated (*Figure 1d*; *Supplementary file 1B*). Consistent with RNA-seq data, despite similar numbers of proteins increasing and decreasing after BRG1 reexpression, those that were upregulated were at a higher Log2FoldChange and significance than those downregulated. Additionally, BRG1 (SMARCA4) was not the most upregulated protein (BRG1/Control Log2FoldChange = 3.04) (*Figure 1d*; *Supplementary file 1B*). CD44 protein level was significantly increased (Log2FoldChange = 2.20), consistent with the increase in gene expression seen with RNA-seq (*Figure 1d*; *Supplementary file 1B*). Looking globally at the genes and proteins identified in both the RNA-seq and proteomics (n = 5528), we found a positive correlation ($R^2$ = 0.407) (*Figure 1—figure supplement 3A*). This correlation was stronger ($R^2$ = 0.774) when filtered for statistically significant changes following BRG1 expression in both assays (RNA-seq: padj <0.05; proteomics: FDR < 0.05; n = 395), suggesting that key gene and protein expression changes are highly correlative in BIN67 cells (*Figure 1—figure supplement 3B*).

Given BRG1's role in chromatin remodeling and the many changes in gene and protein expression following reexpression, we next determined the differences in chromatin accessibility by ATAC-seq. Unsupervised analysis of ATAC-seq samples showed separation based on biological treatment

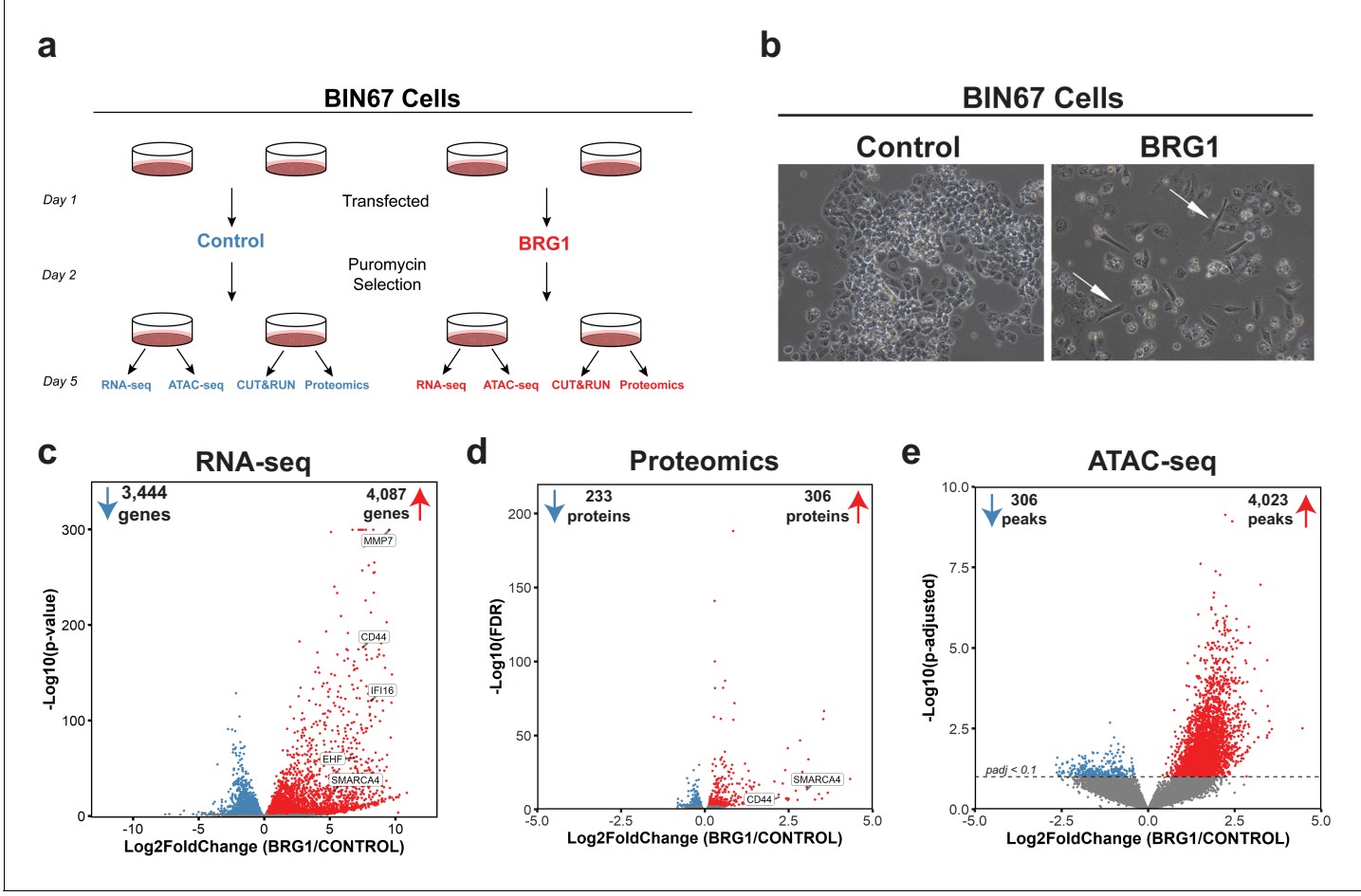

**Figure 1.** BRG1 reexpression induces global morphological, transcriptomic, and epigenetic changes in BIN67 cells. (a) Experimental design for BRG1 reexpression in BIN67 cells and multi-omic data analysis. (b) Following the experimental design in 1a, pictures were taken at 10X phase. BIN67 cells transfected with BRG1 show an elongated morphology (white arrows) relative to control. Additional fields and controls shown in *Figure 1—figure supplement 1*. (c) Volcano plot results of RNA-seq differential gene expression (BRG1/Control) for protein coding genes using DESeq2 (n = 18,507 genes total). Significantly upregulated genes (padj <0.05 and Log2FoldChange > 0) are colored in red (n = 4087 genes). Significantly downregulated genes (padj <0.05 and Log2FoldChange < 0) are colored in blue (n = 3444 genes). Non-significant genes are colored in gray (n = 10,976 genes). SMARCA4/BRG1 and BRG1 target genes are identified. (d) Volcano plot of differential expressed proteins for BIN67 +/- BRG1 reexpression by PECA analysis (n = 5726 total proteins identified). Significantly upregulated proteins (p.fdr <0.05 and Log2FoldChange > 0) are colored in red (n = 306 proteins). Significantly downregulated proteins (p.fdr <0.05 and Log2FoldChange < 0) are colored in blue (n = 233 proteins). Non-significant proteins are identified in gray (n = 5187 proteins). Proteins of SMARCA4/BRG1 and CD44 are identified. (e) Differential peak analysis for ATAC-seq data using DESeq2 (n = 62,308 peaks total). Significantly gained ATAC-seq peaks (padj <0.10 and Log2FoldChange > 0) are identified in red (Gained; n = 4,023). Significantly lost ATAC-seq peaks (padj <0.10 and Log2FoldChange < 0) are identified in blue (Lost; n = 306). Non-significant peaks are identified in Gray (Static/N.S.; n = 57,979). Source data is available in *Figure 1—source data 1*.

The online version of this article includes the following source data and figure supplement(s) for figure 1:

**Source data 1.** Raw data for *Figure 1*.

**Figure supplement 1.** Morphology and protein expression changes in SCCOHT cells following BRG1 reexpression.

**Figure supplement 2.** Unsupervised analysis for multi-omics samples.

**Figure supplement 3.** Correlation of RNA-seq and proteomics data in BIN67 cells.

despite more variation among BRG1 samples (*Figure 1—figure supplement 2C*). Overall, chromatin accessibility globally increased with BRG1 reexpression (*Figure 1e*). Of the 62,308 ATAC-seq peaks identified, 4023 peaks were gained and 306 peaks were lost following BRG1 reexpression (*Figure 1e*). The remaining 57,979 peaks (non-significant/static) did not meet the significance criteria for differential peaks (padj >0.1) (*Figure 1e*).

Taken together, these data show that BRG1 reexpression in BIN67 induces a morphology change along with global upregulation of gene and protein expression and increased chromatin accessibility.

## BRG1 drives an epithelial-like gene, protein, and chromatin signature in SCCOHT cells

Given the morphology change and global changes in gene and protein expression, we questioned whether the cells underwent a change in cellular differentiation after BRG1 reexpression. We first preformed gene set enrichment analysis (GSEA) for Hallmark gene sets in both the RNA-seq (*Figure 2—figure supplement 1A*) and proteomics (*Figure 2—figure supplement 1B*) results. Independently, the RNA-seq and proteomics results showed statistically significant enrichments in many Hallmark pathways (*Figure 2—figure supplement 1A, B*). With different pathways enriched both within and between the RNA-seq and proteomics, this confounded the possibility of a common pathway emerging. In order to answer this issue more specifically, we performed cell type enrichment analysis using xCell (*Aran et al., 2017*) on the RNA-seq gene expression data. Control BIN67 most resembled immune cells such as Th2 cells and mesenchymal stem cells (MSCs) (*Figure 2a*). BIN67 cells reexpressing BRG1 were most like epithelial cells as well as keratinocytes, a specialized epithelial cell (*Figure 2a*). To independently validate the epithelial enrichment in BIN67 cells, we used GSEA for Gene Ontology (GO) terms in both the RNA-seq (*Figure 2b*) and proteomics data (*Figure 2c*). In both data sets, epithelial pathways were significantly enriched in the BRG1 reexpressed cells. We identified GO terms of 'regulation of epithelial cell differentiation' in the gene expression results (*Figure 2b*), and GO terms of 'epithelial cell differentiation' in the proteomics results (*Figure 2c*).

BIN67 cells reexpressing BRG1 showed a strong enrichment for epithelial cell gene and protein signatures. To assess whether the epithelial enrichment was cell line specific, we analyzed RNA-seq data from two other SCCOHT cell lines, SCCOHT-1 and COV434 (*Karnezis et al., 2020*) +/- BRG1 reexpression. Because of the strong enrichment identified by xCell analysis in our data, we looked at all the genes that compose the xCell Epithelial list for gene expression changes following BRG1 reexpression (Log2FoldChange; BRG1/Control) (*Figure 2d*). We included CD44 along with the xCell epithelial genes (n = 131 xCell Epithelial, final n = 132 genes) because of its known role in epithelial cell adhesion (*Sneath and Mangham, 1998*; *Senbanjo and Chellaiah, 2017*) and its established regulation by BRG1 (*Strobeck et al., 2001*; *Reisman et al., 2002*; *Strobeck et al., 2002*). As expected from the xCell results for BIN67, a majority (60%) of the genes in the xCell epithelial list were upregulated in all three SCCOHT cell lines after BRG1 reexpression (n = 79/132 genes - log2FoldChange BRG1/Control > 0) (*Figure 2d*). Furthermore, this was consistent across other previously published BIN67 +/- BRG1 RNA-seq data sets (102/132 genes [*Pan et al., 2019*] and 107/132 genes *Xue et al., 2019b*) and SCCOHT-1+/- BRG1 RNA-seq data set (*Xue et al., 2019b*) (94/132 genes – log2FoldChange BRG1/Control) that all displayed an epithelial-like enrichment following xCell analysis (*Figure 2—figure supplement 1C, D, E, F*, *Supplementary file 4*).

While the xCell epithelial gene list is composed of many epithelial cell genes, it does not fully encompass all the epithelial genes contained in the literature. To validate the xCell results, we independently looked at additional epithelial cell genes found in the literature that were not described by xCell and those used routinely in clinical diagnostics for marking epithelial cells. This list of epithelial cell markers consisted of E-cadherin, desmoplakin, occludin, ZO-1, EpCAM, and various keratins (*Chae et al., 2018*; *Gibbons and Creighton, 2018*; *Lamouille et al., 2014*; *Zeisberg and Neilson, 2009*; *Huang et al., 2018*; *Werner et al., 2020*; *Painter et al., 2010*). Despite a few cell line differences, such as E-cadherin expression, we found that BIN67, SCCOHT-1, and COV434 cells reexpressing BRG1 upregulated the vast majority of the keratin genes (*Figure 2e*, *Supplementary file 4*). The upregulation of keratins in BIN67 and SCCOHT-1 cells following BRG1 reexpression was consistent with previously published data sets (*Figure 2—figure supplement 1G*).

In order to better understand the epithelial signature collectively, we combined the gene list for epithelial cells in xCell (n = 132) and known epithelial markers (n = 42) to curate an epithelial signature (n = 171) to be used for all future analyses. Using rank-based statistics on a single-sample basis (R package singscore), we scored gene expression data from BIN67, SCCOHT-1, and COV434 for enrichment of the curated epithelial signature. Consistent with our previous findings, the epithelial signature score was significantly increased after BRG1 reexpression in all three SCCOHT cell lines

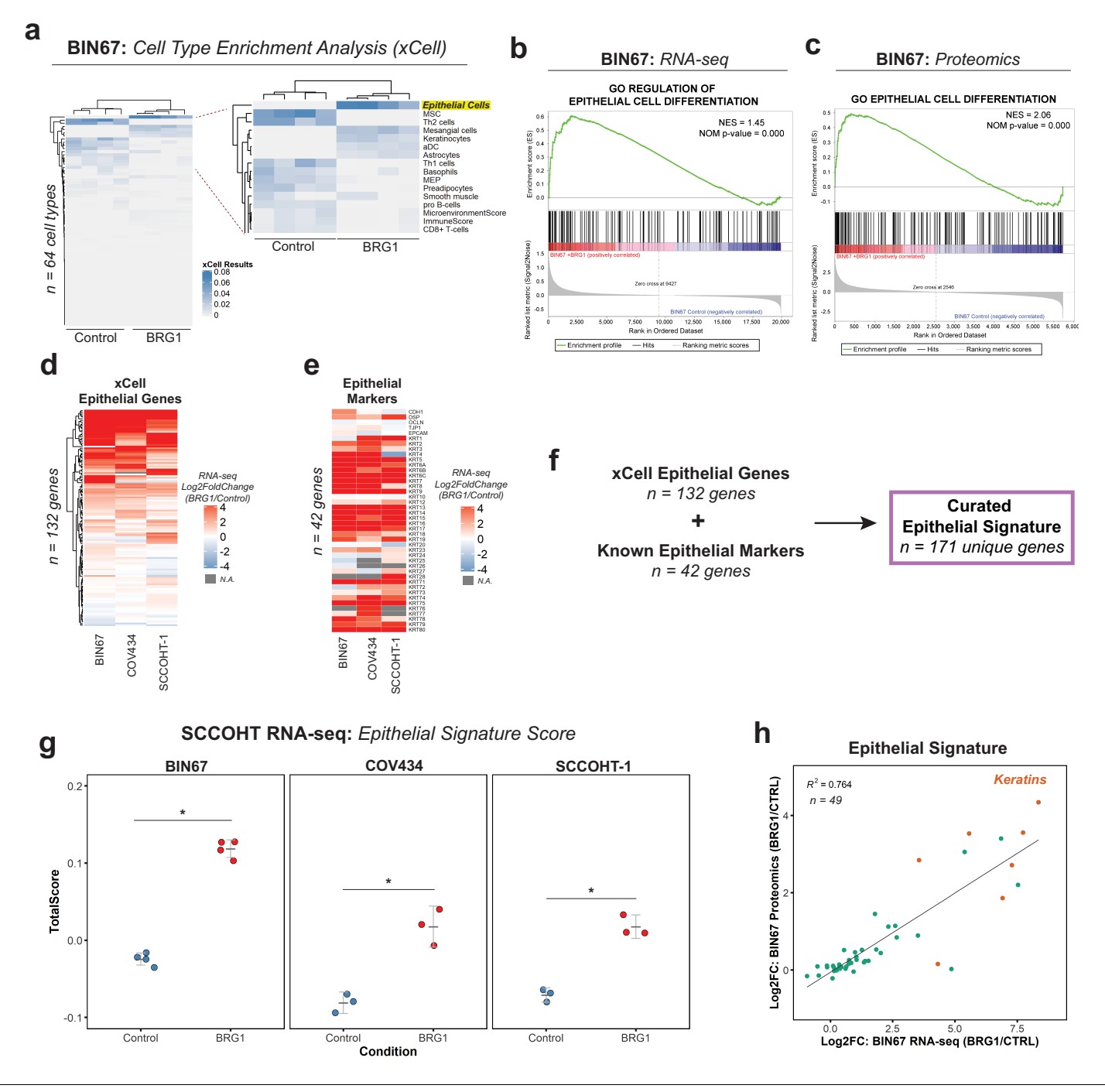

**Figure 2.** BRG1 reexpression induces an epithelial-like expression signature and chromatin profile. (**a**) Heatmap of xCell results from RNA-seq data (gene and library scaled TPMs for input). Epithelial cell category highlighted in yellow. Rows and columns clustered by Pearsons correlation. (**b**) Enrichment plots for epithelial related Gene Ontology (GO) terms in RNA-seq data (VST normalized TPMs) using Gene Set Enrichment Analysis (GSEA; MSigDB GO gene set - C5 all v6.0). (**c**) Enrichment plots for epithelial related GO terms in proteomics data (log2 transformed) using GSEA (MSigDB GO gene set - C5 all v6.0). (**d**) Heatmap of RNA-seq Log2FoldChange (BRG1/Control) for xCell Epithelial Gene set (n = 132) in BIN67, SCCOHT-1, and COV434 cells clustered by Euclidean correlation. (**e**) Heatmap of RNA-seq Log2FoldChange (BRG1/Control) for epithelial cell markers in BIN67, SCCOHT-1, and COV434 cells clustered by Euclidean correlation. For genes that no measurable Log2FoldChange was calculated by DESeq2 are colored in gray. (**f**) Definition of the curated epithelial gene signature. (**g**) Scoring of the epithelial signature in RNA-seq data from SCCOHT cell lines using singscore. (**h**) Scatterplot of gene expression change and associated protein expression changes in BIN67 cells for genes and proteins identified in common from the epithelial signature. Source data is available in *Figure 2—source data 1*.

*Figure 2 continued on next page*

*Figure 2 continued*

The online version of this article includes the following source data and figure supplement(s) for figure 2:

**Source data 1.** Raw data for *Figure 2*.
**Figure supplement 1.** Epithelial genes are upregulated following BRG1 expression in SCCOHT cell lines.
**Figure supplement 2.** Epithelial signature scoring in previously published SCCOHT datasets.

(Welch Two Sample t-test p-values: BIN67 = 2.988e-06, COV434 = 0.0075, and SCCOHT-1 = 0.0014) (*Figure 2g*). The epithelial signature score additionally increased in all previously published data sets for BIN67 and SCCOHT-1 (*Figure 2—figure supplement 2A, B, C*).

We found gene and protein expression was strongly correlative in BIN67 cells (*Figure 1—figure supplement 3A, B*). Therefore, we questioned if the genes and proteins identified in from the epithelial signature correlated following BRG1 reexpression in BIN67 cells. Of the 171 unique genes in the epithelial signature, 49 were found to have measurable gene expression and protein changes (*Figure 2h*). These genes and proteins were found to be strongly correlated ($R^2$ = 0.764) with the keratins being among the most highly upregulated in both gene and protein levels (*Figure 2h*). Overall, we concluded that the epithelial signature was upregulated after BRG1 expression in SCCOHT cell lines both at gene and protein expression level.

BRG1's main role is to provide the energy to remodel the chromatin landscape through its ATPase activity. To understand if the enrichment of the epithelial signature in BIN67 was dependent on the ATPase activity, we utilized previously published RNA-seq data that expressed BRG1 ATPase mutants in BIN67 (*Pan et al., 2019*). We found that the expression of both a partial (T910M) or complete (K785R) loss of catalytic activity for BRG1 in BIN67 did not induce the epithelial signature (*Figure 2—figure supplement 2C*). This suggests that the induction of the epithelial gene signature by BRG1 expression in SCCOHT cells is dependent upon its catalytic activity.

## BRG1-induced epithelial signature is broadly applicable to other tumor types and primary SCCOHT tumors

Other adult cancers, such as NSCLC, contain mutations in BRG1 but have a higher genetic burden than SCCOHT tumors. In order to understand if the enrichment of the epithelial signature is SCCOHT specific or is applicable to other BRG1-deficient tumors, we analyzed RNA-seq data from two NSCLC cell lines. We first looked at A427 +/- BRG1 because similar to SCCOHT cell lines, A427 lacks BRM expression. Following BRG1 expression, A427 cells significantly increased enrichment for the epithelial signature (Welch Two Sample t-test p-value=0.00067) (*Figure 3a*). To confirm these results, we used RNA-seq data we recently reported for H358 cells with knockout of BRG1 or BRM expression (*Song et al., 2020*). H358 cells (Parent) express endogenous levels of BRG1 and BRM and score high for the epithelial signature (*Figure 3b*). However, when BRG1 expression is lost, the score for the epithelial signature significantly decreased compared to the parent cell line (Welch Two Sample t-test p-value = 0.04437; *Figure 3b*). In contrast, the loss of BRM expression did not decrease the epithelial signature, rather it slightly increased it compared to the parent cell line (Welch Two Sample t-test p-value=0.02038; *Figure 3b*). Together, these data indicate that the BRG1-dependent induction of the epithelial signature appears broadly applicable to other BRG1 mutant tumors like NSCLC.

Because the transcriptomic and proteomic profiles after BRG1 expression in BIN67 resembled an epithelial-like cell, we questioned if the chromatin landscape gained after BRG1 expression resembled those seen in tumors originating in epithelial tissues. To answer this, we compared our chromatin accessibility profiles in the presence of BRG1 to those tumor types in The Cancer Genome Atlas (TCGA) ATAC-seq data set (n = 23 tumor types) (*Corces et al., 2018*). To measure which tumor types were most similar to epithelial cells, we used previously calculated xCell scores in the epithelial cell category for TCGA tumors (*Aran et al., 2017*) (n = 9358 tumors) and averaged it based on each tumor type to have a mean xCell epithelial score for a given tumor type (n = 33 tumor types) (*Aran et al., 2017*). In order to compare chromatin profiles, we determined the percentage of peaks in each tumor type that overlapped with the gained ATAC-seq peaks (n = 4023 peaks). Tumor types that had both ATAC-seq data and a calculated xCell score were used for further analysis. Percentage of ATAC-seq peaks that overlapped with TCGA ATAC-seq peaks ranged from 19.6% (TGCT -

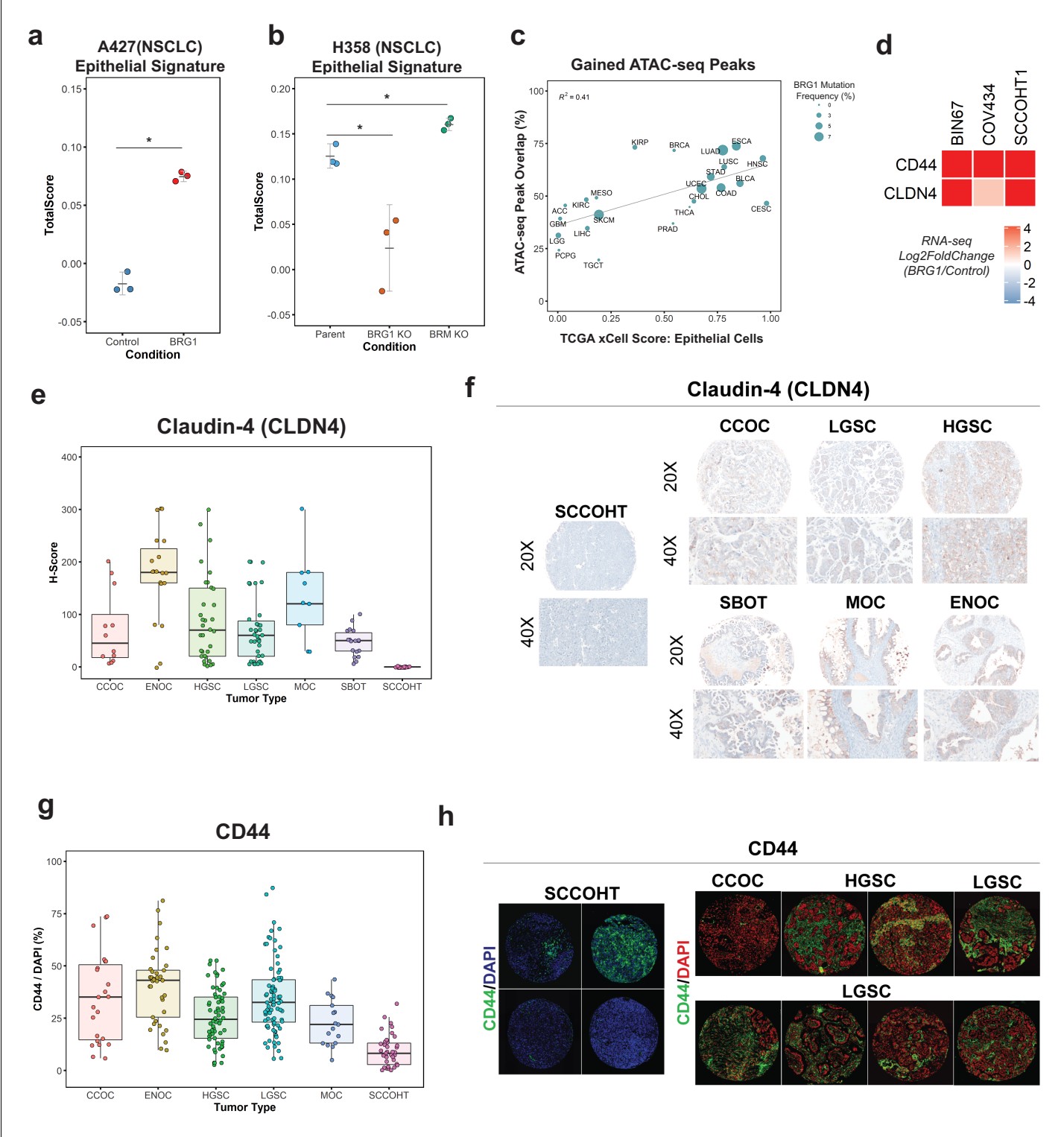

**Figure 3.** Loss of the epithelial signature is broadly applicable to BRG1-deficient NSCLC cell lines and SCCOHT primary tumors. (**a, b**) Scoring of epithelial signature in RNA-seq data from NSCLC cell lines +/- BRG1 expression (A427) and +/- BRG1/BRM knockout (H358). (**C**) Scatterplot of percent peak overlap of gained ATAC-seq peaks with TCGA ATAC-seq peaks versus the averaged xCell calculated epithelial cell score for each TCGA tumor type. The frequency of BRG1 mutations identified per tumor type (reported as a percentage) is indicated by the size of the point. (**d**) Heatmap of CD44 and Claudin-4 (CLDN4) gene expression changes following BRG1 expression in SCCOHT cell lines. (**e, f**) Immunohistochemistry scoring and representative images for Claudin-4 expression from a TMA containing gynecological tumors. (**g, h**) Immunofluorescence scoring and representative

*Figure 3 continued on next page*

*Figure 3 continued*

images for CD44 expression at ×20 magnification from a TMA containing gynecological tumors. In both Claudin-4 and CD44 staining (**e–h**), multiple core samples (between 2 and 6 cores) were taken per tumor and reported as single points in boxplots. Total numbers reported as the number of cores taken per tumor type. Small Cell Carcinoma of the Ovary, Hypercalcemic Type (SCCOHT; n = 16 cores for CLDN4, n = 39 cores for CD44). Clear Cell Ovarian Carcinoma (CCOC; n = 12 cores for CLDN4, n = 23 cores for CD44). High-Grade Serous Ovarian Carcinoma (HGSC; n = 33 cores for CLDN4, n = 64 cores for CD44). Low-Grade Serous Ovarian Carcinoma (LGSC; n = 38 cores for CLDN4, n = 83 cores for CD44). Endometriod Ovarian Carcinoma (ENOC; n = 19 cores for CLDN4, n = 37 cores for CD44). Mucinous Ovarian Carcinoma (MOC; n = 9 cores for CLDN4, n = 17 cores for CD44). Serous Borderline Ovarian Tumor, pre-cursor legion of LGSC (SBOT; n = 18 cores for CLDN4; CD44 staining not performed for this tumor type). Source data is available in *Figure 3—source data 1*.

The online version of this article includes the following source data for figure 3:

**Source data 1.** Raw data for *Figure 3*.

Testicular Germ Cell Tumors) to 73.7% (ESCA - Esophageal carcinoma) (*Figure 3c*). Average scores for the epithelial category ranged from 0.0025 (LGG – Low-Grade Gliomas) to 0.982 (CESC - Cervical squamous cell carcinoma and endocervical adenocarcinoma) (*Figure 3c*). We found a positive correlation ($R^2$ = 0.41) between percentage of gained ATAC-seq peaks that overlapped with xCell Epithelial score that was statistically significant (Pearson's correlation p-value=0.00095) (*Figure 3c*). We looked further to the frequency of BRG1 mutations in these tumor types; however, there was not a strong correlation between BRG1 mutation frequency and loss of xCell Epithelial score, likely due to tumor heterogeneity and the lower overall frequency of BRG1 loss as compared to SCCOHT BRG1 mutation frequency (*Figure 3c*). These results indicate that the altered chromatin landscape following BRG1 reexpression in BIN67 cells is similar to those of epithelial originating tumors, consistent with our results of epithelial-like gene and protein expression.

To validate the epithelial signature induction by BRG1 in SCCOHT cell lines, we chose two epithelial targets (CD44 and Claudin-4/CLDN4) that increased in gene expression in all three SCCOHT cell lines (*Figure 3d*) and additionally increased protein expression levels following BRG1 in BIN67 and SCCOHT-1 (*Figure 1—figure supplement 1B, C*). CD44 is a well-characterized cell surface receptor with several isoforms expressed in epithelial cells (*Sneath and Mangham, 1998*; *Senbanjo and Chellaiah, 2017*). CD44 has also been shown as a BRG1 target gene (*Strobeck et al., 2001*; *Reisman et al., 2002*; *Strobeck et al., 2002*), making it another excellent candidate for further validation studies. Claudin-4 (*CLDN4*) is involved in epithelial tight junctions and recently described as a marker of epithelial origin by IHC (*Schaefer et al., 2017*; *Tessier-Cloutier et al., 2018*). Thus, we used CD44 and Claudin-4 for further analysis to determine if these epithelial markers were lowly expressed in primary SCCOHT tumors as we had hypothesized. A tissue microarray of primary tumors from SCCOHT and other gynecological cancers were stained for Claudin-4 and CD44 (*Figure 3e,f,g,h*). SCCOHT primary tumors lacked expression of Claudin-4 (average H-Score = 0), whereas High-Grade Serous Ovarian Cancer (HGSC) tumors had a dynamic range of expression with some expressing very strongly (*Figure 3e,f*). Consistent with Claudin-4 staining, SCCOHT primary tumors scored the lowest overall for CD44 expression (average CD44/DAPI: SCCOHT = 9.43%, CCOC = 34.7%, HGSC = 25.7%, LGSC = 35%, and MOC = 23.0%) (*Figure 3g*). The range of CD44 expression in primary SCCOHT tumors was more dynamic than with Claudin-4, shown by select tumor cores that moderately expressed CD44 (range of CD44/DAPI in SCCOHT = 0.22–31.9%) (*Figure 3h*). Other gynecological tumors on average had a higher expression level of CD44 than SCCOHT primary tumors (*Figure 3g,h*). Therefore, SCCOHT primary tumors lack the expression of select epithelial markers, consistent with our findings in SCCOHT cell lines in the absence of BRG1 expression.

## AP-1 motifs are enriched in BRG1 recruited and gained chromatin accessibility

BRG1 functions by altering nucleosome occupancy in a way that affects gene expression and other DNA-mediated processes. Chromatin accessibility that was gained following BRG1 reexpression in BIN67 cells was most similar to epithelial originating tumors, leading us to question if BRG1 was directly affecting epithelial gene programs. We next identified where in the genome chromatin was altered and which of those sites were direct targets of BRG1. After BRG1 reexpression, chromatin accessibility was gained mostly at introns and intergenic regions (*Figure 4A*). A search for a common

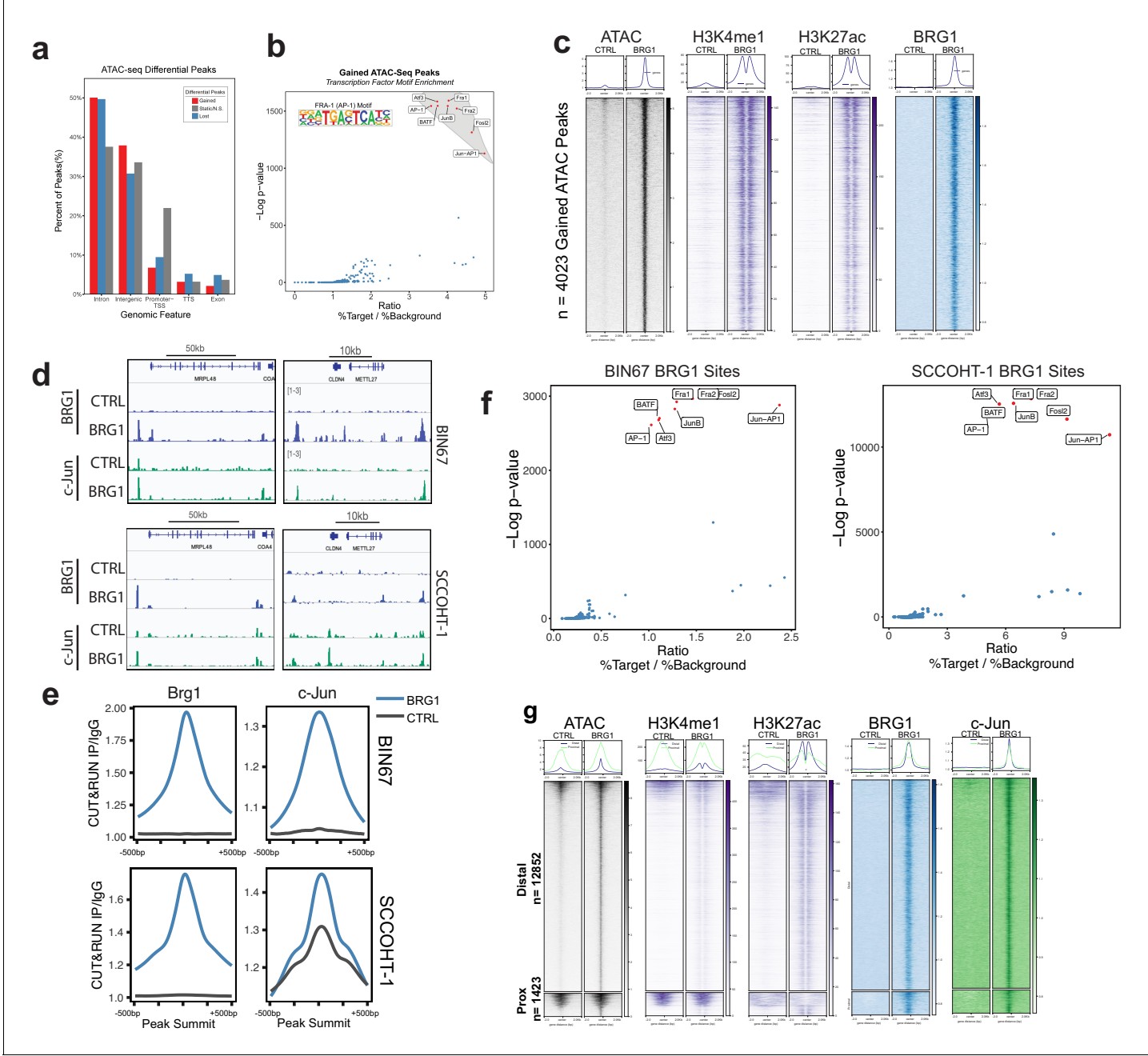

**Figure 4.** AP-1 motifs are enriched at BRG1 recruited chromatin and gained chromatin accessibility. (a) Genomic annotations by HOMER for differential ATAC-seq peaks on standard chromosomes (Gained: n = 4023 peaks; Static/N.S.: n = 57,747 peaks; and Lost: n = 306 peaks). (b) Known transcription factor motif analysis for ATAC-seq gained peaks (n = 4023 peaks) performed by HOMER with Static/N.S. ATAC-seq peaks used as background. AP-1 family members indicated by red points and highlighted in gray. (c) Heatmap of ATAC-seq signal, H3K27Ac signal, and H3K4me1 signal, and BRG1 signal in CUT and RUN experiments +/- BRG1 reexpression at the gained ATAC-seq peaks in BIN67 cells. (d) Examples of browser tracks showing enrichment (IP/IgG) of CUT and RUN data for BRG1 and Jun in BIN67 and SCCOHT-1 cells. (e) Metaplots showing BRG1 or c-Jun signal relative to Brg1 peaks (+/- 500 bp) in BIN67 or HT1 cells comparing Control or BRG1 reexpressed cells. (f) Known transcription factor motif analysis for BRG1 peaks identified in BIN67 or SCCOHT-1 cells reexpressing BRG1 (n = 16,283, 31,379). (g) ATAC-seq signal, H3K27ac, H3K4me1 ChIP-seq signal BRG1 and c-Jun CUT and RUN signal (+/- BRG1 reexpression) at FRA1 motifs contained in open chromatin in the genome. Promoter regions defined as +/- 1 kb of TSS (n = 1423 motif sites) and all remaining sites defined as distal (n = 12,852 motif sites). Heatmaps in all panels are in the same sorted order. Source data is available in *Figure 4—source data 1*.

The online version of this article includes the following source data and figure supplement(s) for figure 4:

**Source data 1.** Raw data for *Figure 4*.

*Figure 4 continued on next page*

*Figure 4 continued*

**Figure supplement 1.** c-Jun is redistributed in SCCOHT-1 following BRG1 reexpression.
**Figure supplement 1—source data 1.** Raw data for *Figure 4—figure supplement 1*.

motif present in sites with increased chromatin accessibility showed a strong enrichment for various AP-1 family members (*Figure 4B*). To determine if the gained distal regions are enhancers, we utilized previously published ChIP-seq data for enhancer marks in BIN67 +/- BRG1 reexpression (*Pan et al., 2019*). We found that the gained accessible regions in BIN67 additionally gained H3K27ac and H3K4me1 enhancer marks, defining them as enhancer regions (*Figure 4C*; *Pan et al., 2019*). This is consistent with previous reports of BRG1 reexpression in NSCLC (*Xue et al., 2019a*) and SCCOHT (BIN67) (*Pan et al., 2019*). While distal regions changed in chromatin accessibility, the promoter regions were largely composed of static ATAC-seq peaks, representing accessible chromatin both in the presence and absence of BRG1 in BIN67 cells (*Figure 4A*).

We next questioned if the locations of changing chromatin accessibility were consistent with BRG1 recruitment. Using CUT and RUN (*Skene and Henikoff, 2017*), we found that BRG1 was recruited to sites of increased chromatin accessibility (*Figure 4C*, *Supplementary file 5*, *Supplementary file 6*). Given the transcription factor motif analysis on gained ATAC-seq peaks showed a strong enrichment for various AP-1 family members, we asked whether the AP-1 member c-Jun was recruited to sites with increased that increase chromatin accessibility. CUT-and-RUN data for c-Jun showed strong enrichment at these sites (*Figure 4C, D*). Notably, these regions were generally not previously bound by c-Jun in BIN67 cells. However, we noted very little c-Jun binding in BIN67 prior to BRG1 reexpression in general (*Figure 4C,D*). In contrast, while c-Jun bound well prior to BRG1 reexpression in HT1 cells, there was a significant amount of c-Jun relocalization following BRG1 (*Figure 4—figure supplement 1a*) Genes that newly recruited both c-Jun and BRG1 increased expression, while those that lost c-Jun and did not gain BRG1 expression decreased expression (*Figure 4—figure supplement 1b*). Notably c-Jun expression was uniformly high in SCCOHT-1 cells, whereas in BIN67 c-Jun was strongly induced by BRG1 reexpression (*Figure 4—figure supplement 1c*).

Next, we searched for enriched motifs in BRG1-bound regions in both BIN67 and SCCOHT-1 cells in which BRG1 has been reexpressed. In both cases, motifs for AP-1 family members segregated from other transcription factor motifs based on significance (p-value) and ratio of sites containing a motif over background (*Figure 4F*), consistent with the ATAC-seq data (*Figure 4E*; *Supplementary file 2B*, *Supplementary file 7*).

Since the AP-1 motif was common among the ATAC-seq gained peaks and BRG1 peaks, we next assessed whether BRG1 was recruited to AP-1 (FRA1) motifs found in accessible chromatin. We assigned each FRA1 motif contained within open chromatin to either promoter (+/- 1 kb TSS) or distal region and analyzed BRG1 localization at these classes. At promoter regions, ATAC-seq signal mostly remained open with a small number of sites opening after BRG1 reexpression (*Figure 4G*). However, at distal regions containing a FRA1 site, nearly all locations gained chromatin accessibility and enhancer marks (H3K27ac and H3K4me1) after BRG1 reexpression (*Figure 4G*). For both promoter and distal regions, BRG1 was recruited to FRA1 locations (*Figure 4G*). AP-1 family member c-Jun was also strongly recruited to its motifs present in both promoter proximal and distal open chromatin (*Figure 4g*). Consistent with previous reports of AP-1 and SWI/SNF interactions (*Vierbuchen et al., 2017*), these results suggested that AP-1 is a common motif associated with BRG1 both at promoter and distal locations, implicating AP-1 transcriptional control as an important mediator of SCCOHT development.

## Epithelial-like gene signature in SCCOHT is driven by AP-1 mechanism

Because AP-1 motifs were highly enriched in sites that increase chromatin accessibility and recruit BRG1, we questioned if the epithelial-like signature observed after BRG1 reexpression is driven through AP-1-dependent mechanisms. We first looked at the distribution of expression changes for the epithelial genes to determine if a particular subset of epithelial genes were driving the epithelial signature (*Figure 2*). We found a bimodal distribution of expression changes (*Figure 5A*). This was observed in both BIN67and SCCOHT-1 cells with a strong concordance in the expression levels of

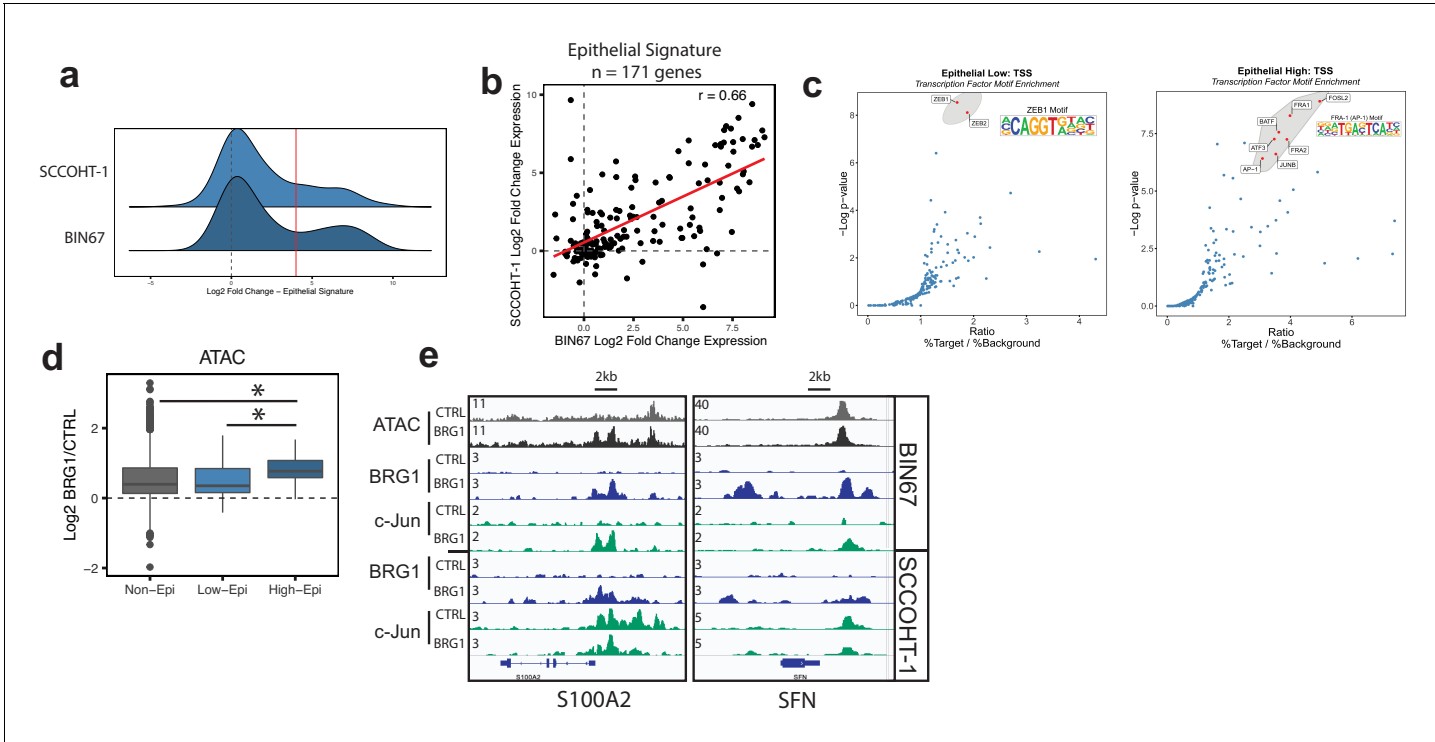

**Figure 5.** Epithelial genes dependent on AP-1 are highly upregulated by BRG1 expression. (a) Density plot of RNA-seq Log2FoldChange (BRG1/Control) for epithelial signature genes (n = 171 genes). Red dotted line indicates cutoff point for 'High' category (Log2FoldChange > 4). (b) Scatterplot of log2FoldChanges (+/- BRG1) in BIN67 (x-axis) and SCCOHT-1(y-axis). Pearson's correlation. = 0.66. (c) Known transcription factor motif analysis at the promoter of (b) low epithelial gene and (c) high epithelial genes from BIN67 xCell gene set. Default settings defined by HOMER were used for promoter region (−300 bp, +50 bp from Transcription Start Site, TSS) and background. Transcription factors that were also expressed in RNA-seq data were plotted. Transcription factors that were not expressed at a mean count across all samples greater than four TPMs were eliminated. Highest ranking based on -log(p-value) and Ratio (% Target / % Background) and of similar TF families plotted in red and encircled in gray. (d) Box plot of ATAC-seq signal (Log2FoldChange; BRG1/Control) at the BRG1 CUT and RUN peaks annotated to all high epithelial genes (n = 48 peaks), all low epithelial genes (n = 94 peaks) and non-epithelial genes (n = 15,562 peaks). Non-epithelial genes defined as a BRG1 CUT and RUN-seq peak annotated to nearest gene not contained in either high or low group. Statistical analysis was performed by Wilcoxon Rank Sum Test on low epithelial versus non-epithelial (N.S., p-value=0.907), low epithelial versus high epithelial (p-value=3.5e−4), and high epithelial versus non-epithelial (p-value=5.4e−5). (e) IGV browser tracks of two epithelial signature loci. Source data is available in *Figure 5—source data 1*.

The online version of this article includes the following source data and figure supplement(s) for figure 5:

**Source data 1.** Raw data for *Figure 5*.
**Figure supplement 1.** Transcription factor motif analysis at promoter of xCell gene signatures.

the epithelial signature in both cell lines (Pearson's r = 0.66, p-value=3.95e−22, *Figure 5B*). We hypothesized that at least two different transcription factors may drive the epithelial expression, one driving the epithelial low group (n = 100 genes BIN67) and one driving the epithelial high group (n = 48 BIN67, 40 SCCOHT-1, Log2FoldChange > 4, n = 30 found in both) (*Figure 5C*). We performed transcription factor motif analysis at the promoter regions of the epithelial low and high groups separately as defined by BIN67 for the xCell Genes (n = 100 low, n = 32 high). The epithelial low group was strongly enriched in ZEB1/2 transcription factor motifs, not surprising given their well-characterized role in regulating EMT/epithelial cells (*Lamouille et al., 2014*; *Figure 5C*). However, the epithelial high group showed a strong enrichment in AP-1 family members (*Figure 5C*). AP-1 motifs were also found in the epithelial low group, although not to the same level of enrichment in the epithelial high group (*Figure 5—figure supplement 1A*). Other than keratinocytes, which is a highly specialized epithelial cell, AP-1 motifs were less enriched in the promoters of genes that define other cell types from xCell (*Figure 5—figure supplement 1*). Thus, we hypothesized that increased expression of epithelial high gene set depended upon the interaction between BRG1 and AP-1 in BIN67 cells.

The recruitment and co-localization of BRG1 and c-Jun to the chromatin lead us to question if chromatin accessibility was differentially changing at these sites. ATAC-seq signal (log2FoldChange; BRG1/Control) was determined at BRG1 CUT and RUN peaks in BIN67 that are annotated to epithelial low, epithelial high, or non-epithelial genes (*Figure 5d*). Chromatin accessibility increased more for the high epithelial group than for the low epithelial group and non-epithelial groups (*Figure 5d*). At individual loci, strong recruitment of both BRG1 and c-Jun can be seen near the promoters and in upstream regions of genes (*Figure 5d*, S100A2 and SFN). These results collectively support the concept that the increased expression of the epithelial high genes in BIN67 depends upon BRG1 recruitment and chromatin remodeling combined with the activity of AP-1.

## AP-1 activity is necessary for BRG1-dependent epithelial gene and protein expression

We next asked if AP-1 activity was biologically necessary for the BRG1-induced epithelial-like signature. Because a variety of AP-1 families and members exist, we first in BIN67 and SCCOHT-1 to determine if the expression of a particular AP-1 family predominates across SCCOHT cell lines (*Figure 6—figure supplement 1*). We found that the gene expression of many members corresponding to each of the four main families (FOS, JUN, ATF, MAF) were upregulated after BRG1 reexpression in BIN67 and SCCOHT-1 cells (*Figure 6—figure supplement 1A, B*). In order to inactivate a broad range of these AP-1 family members, we created an inducible dominant negative AP-1 cell line in BIN67 cells (BIN67 pIND20-FLAG-A-FOS). Unlike other dominate negative AP-1 constructs that specifically delete the transactivation domain in particular family members and still allow for DNA binding (*Tichelaar et al., 2010*), A-FOS prevents the activity of AP-1 by an acidic extension that closes the DNA-binding domain, preventing it from binding and localizing on DNA (*Olive et al., 1997*). Therefore, A-FOS interacts ubiquitously with any AP-1 family member, allowing us to address if the binding of AP-1 to DNA is necessary along with its activity, without focusing on a particular family member/dimer. BIN67 pIND20-FLAG-A-FOS cell lines were first characterized for optimal A-FOS induction (*Figure 6—figure supplement 2A*). Cells were plated and induced with doxycycline (DOX) for 2 days and then processed with the BRG1 transfection-fast puromycin selection as described previously, and then harvested for protein extraction and RNA-seq (*Figure 6a*).

Reexpression of BRG1 in BIN67 pIND20-FLAG-A-FOS cells (-DOX conditions) resulted in 767 genes that differentially changed (687 significantly upregulated, 80 significantly downregulated) (*Figure 6—figure supplement 2B*; *Supplementary file 3A*). The log2FoldChange for these 767 genes were plotted against BIN67 parent to determine if there were any changes in gene expression due to the presence of the inducible vector (*Figure 6—figure supplement 2B*). We found that BIN67 pIND20-FLAG-A-FOS (-DOX condition; absence of A-FOS) cells +/- BRG1 recapitulated the parental BIN67 +/- BRG1 by a strong correlation ($R^2 = 0.724$, Pearson's Correlation p-value<2.2e-16) in log2-FoldChange for the 767 statistically significant genes (*Figure 6—figure supplement 2B*). To determine the global effect that A-FOS induction and BRG1 reexpression have in BIN67 cells, we performed unsupervised analysis (PCA) for RNA-seq samples (*Figure 6b*). Unsupervised principal component analysis of the four conditions (+/- BRG1, +/- DOX(A-FOS)) in BIN67 pIND20-FLAG-A-FOS showed that that all control transfected conditions clustered together (blue), regardless of the induction of A-FOS (*Figure 6b*). However, BRG1 transfected conditions segregated separately based on the presence or absence of DOX (A-FOS) (red), in addition to segregating from control transfected samples (*Figure 6b*). The little to no effect that the induction in of A-FOS had on control transfected cells was confirmed with differential expression analysis of RNA-seq data where only eight genes were significantly upregulated and 24 genes were significantly downregulated (*Figure 6—figure supplement 2C*; *Supplementary file 3B*). This is a surprising result, as we would have expected that blocking AP-1 binding activity would have significant effects on gene expression given AP-1's extensive role in cell cycle and regulation of other cellular processes (*Chinenov and Kerppola, 2001*; *Kappelmann et al., 2014*; *Hess et al., 2004*; *Shaulian and Karin, 2001*; *Shaulian and Karin, 2002*). However, as expected with the PCA analysis, the induction of A-FOS had a greater effect on BRG1 reexpressed cells (*Figure 6—figure supplement 2D*). We found that more genes were downregulated upon A-FOS induction than upregulated when BRG1 was reexpressed (405 significantly downregulated and 52 significantly upregulated) (*Figure 6—figure supplement 2D*; *Supplementary file 3C*). Of the 405 genes that were downregulated in the presence of A-FOS (+DOX) and BRG1 reexpression, nearly all these genes were upregulated by BRG1 in BIN67 parental

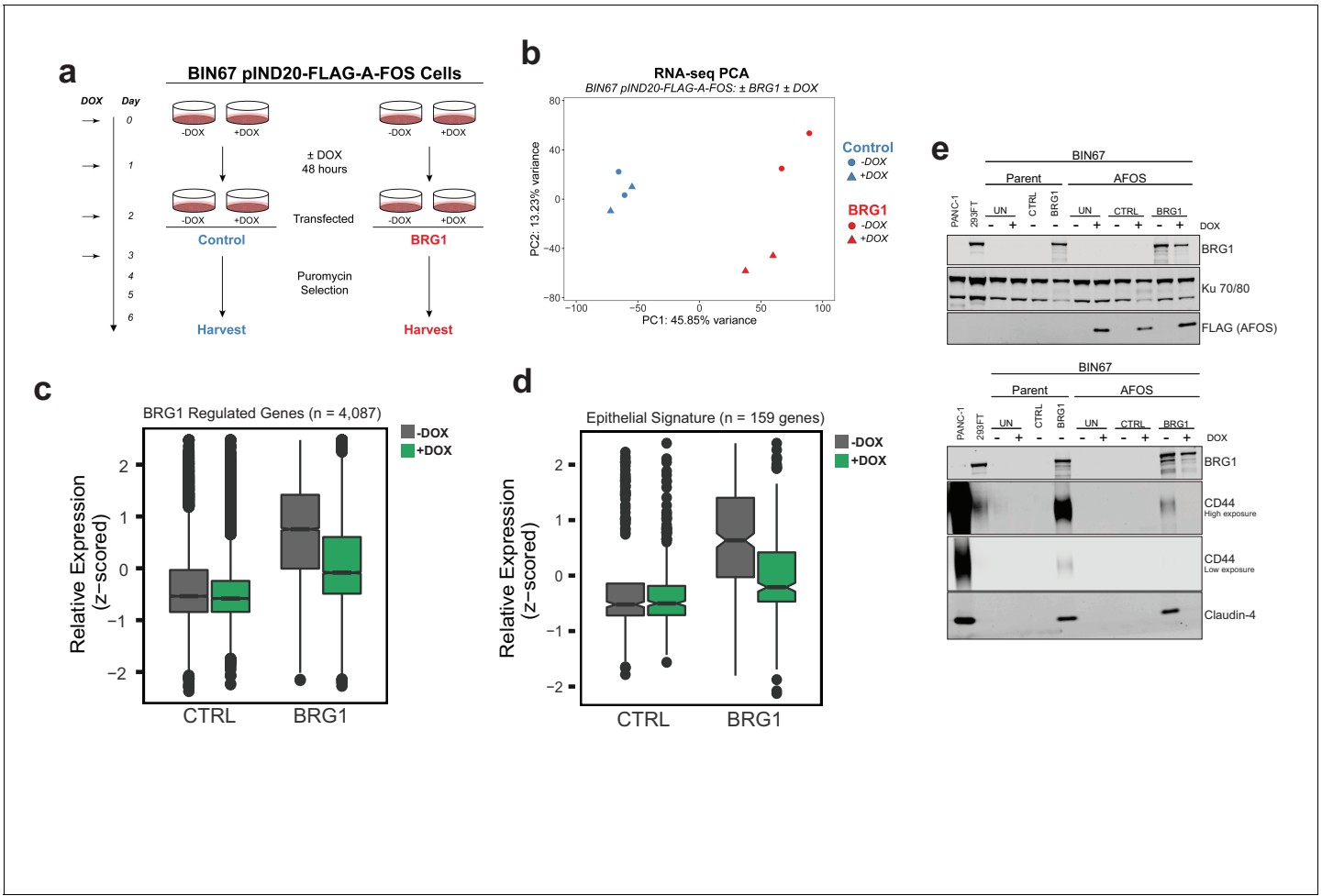

**Figure 6.** AP-1 activity is necessary for BRG1 induction of epithelial targets. (**a**) Experimental design for BRG1 reexpression and A-FOS induction. BIN67 pIND20-FLAG-A-FOS cells were induced for 2 days with DOX prior to BRG1 reexpression. BRG1 reexpression followed previous reexpression studies (*Figure 1a*). (**b**) PCA analysis of BIN67 pIND20-FLAG-A-FOS (+/- DOX, +/- BRG1) RNA-seq results. (**c**) Boxplot of gene expression (z-scored) in BIN67 pIND20-FLAG-A-FOS (+/- DOX, +/- BRG1) for the significantly increased genes identified by BRG1 expression in parent BIN67 (*Figure 1C*; n = 4087), p-values ~0 between Control samples and 2.4e−6 between BRG1 samples by (Wilcoxon signed-rank test). (**d**) Boxplot of gene expression (z-scored) in BIN67 pIND20-FLAG-A-FOS (+/- DOX, +/- BRG1) for the genes in the epithelial signature that were measurable by RNA-seq n = 159. (p-values 0.96 between Control samples, and 4.95e−22 between BRG1 samples by Wilcoxon signed-rank test). (**e**) Western blots for BIN67 +/- BRG1 +/- AFOS following experimental design in (**a**). CD44 shown at low/high exposure by LiCOR detection. Blots are technical duplicates of the same protein sample run in parallel. Par = Parent. AFOS = BIN67 pIND20-FLAG-A-FOS cells. CTRL = control transfected. UN = untransfected. PANC-1 serves as a positive control for CD44 and Claudin-4 protein expression. Ku 70/80 serves as an internal loading control. Source data is available in *Figure 6—source data 1*. The online version of this article includes the following source data and figure supplement(s) for figure 6:

**Source data 1.** Raw data for *Figure 6*.
**Figure supplement 1.** Gene expression of AP-1 family members in BIN67 and SCCOHT-1 cells.
**Figure supplement 2.** A-FOS induction in BIN67 pIND20-FLAG-A-FOS cells.

cell line and BIN67 pIND20-FLAG-A-FOS, -DOX conditions (*Figure 6—figure supplement 2E*). Since AFOS induction alone had little effect on BIN67 cells and the greatest changes were seen with both the reexpression of BRG1 and induction of A-FOS, we determined that BRG1 is necessary for nearly all AP-1-dependent gene changes in BIN67 cells.

With the global effect that AP-1 activity loss had on BRG1-dependent targets in BIN67 cells, we first looked at the effect that A-FOS had on the previously identified upregulated BRG1 genes in BIN67 (*Figure 1C*). We found that the induction of A-FOS with BRG1 expression mitigated the increase of many, but not all BRG1-dependent gene expression in BIN67 cells (*Figure 6c*). We next hypothesized that this was also occurring for the epithelial signature due to the enrichment with AP-

1 motifs. We looked at all genes in the epithelial signature that were identified as measurable counts in BIN67 pIND20-AFOS (n = 159 genes). We found a decrease in most expression levels of the epithelial genes when both BRG1 and A-FOS were induced in BIN67, as compared to BRG1 induction alone (*Figure 6D*). Thus, most, but not all BRG1 dependent gene expression changes require AP-1 activity. To validate this, we analyzed the protein expression level of two epithelial markers (CD44 and Claudin-4) +/- BRG1 and +/- A FOS induction (*Figure 6e*). CD44 has also been shown as a BRG1 target gene (*Strobeck et al., 2001*; *Reisman et al., 2002*; *Strobeck et al., 2002*) and an AP-1 target gene (*Lamb et al., 1997*; *Yamamichi-Nishina et al., 2003*), making it an ideal protein for further validation. BRG1 reexpression in BIN67 parental cells and BIN67 pIND20-FLAG-A-FOS (-DOX conditions) induced CD44 protein expression (*Figure 6e*). In the presence of A-FOS and BRG1 expression, CD44 protein expression was absent (*Figure 6e*). We next examined Claudin-4 (*CLDN4*) expression. Following BRG1 reexpression in BIN67 parental cells and BIN67 pIND20-FLAG-A-FOS (-DOX conditions; A-FOS absent), Claudin-4 expression increased protein (*Figure 6e*) levels. However, in the presence of A-FOS and BRG1 expression, Claudin-4 protein expression was lost (*Figure 6e*).

In summary, we observed that global gene expression of epithelial targets induced by BRG1 depended upon the activity of AP-1. We validated this dependence by the loss of protein expression in epithelial target genes when AP-1 activity was lost in the presence of BRG1 reexpression in BIN67 cells. Thus, both active AP-1 and BRG1 reexpression in BIN67 are necessary for the induction of epithelial targets, suggesting that neither alone is sufficient to induce their expression. These results support a model where after BRG1 reexpression, chromatin accessibility is gained at distal enhancer regions enriched with AP-1 motifs accompanied by recruitment of BRG1 (*Figure 7*). Additionally, BRG1 is recruited to the promoters of epithelial genes that contain AP-1 motifs (*Figure 7*). This results in an increased gene and protein expression of epithelial markers. Our results demonstrate that BRG1 and AP-1 together have an important biological impact on inducing an epithelial-like signature in SCCOHT cells.

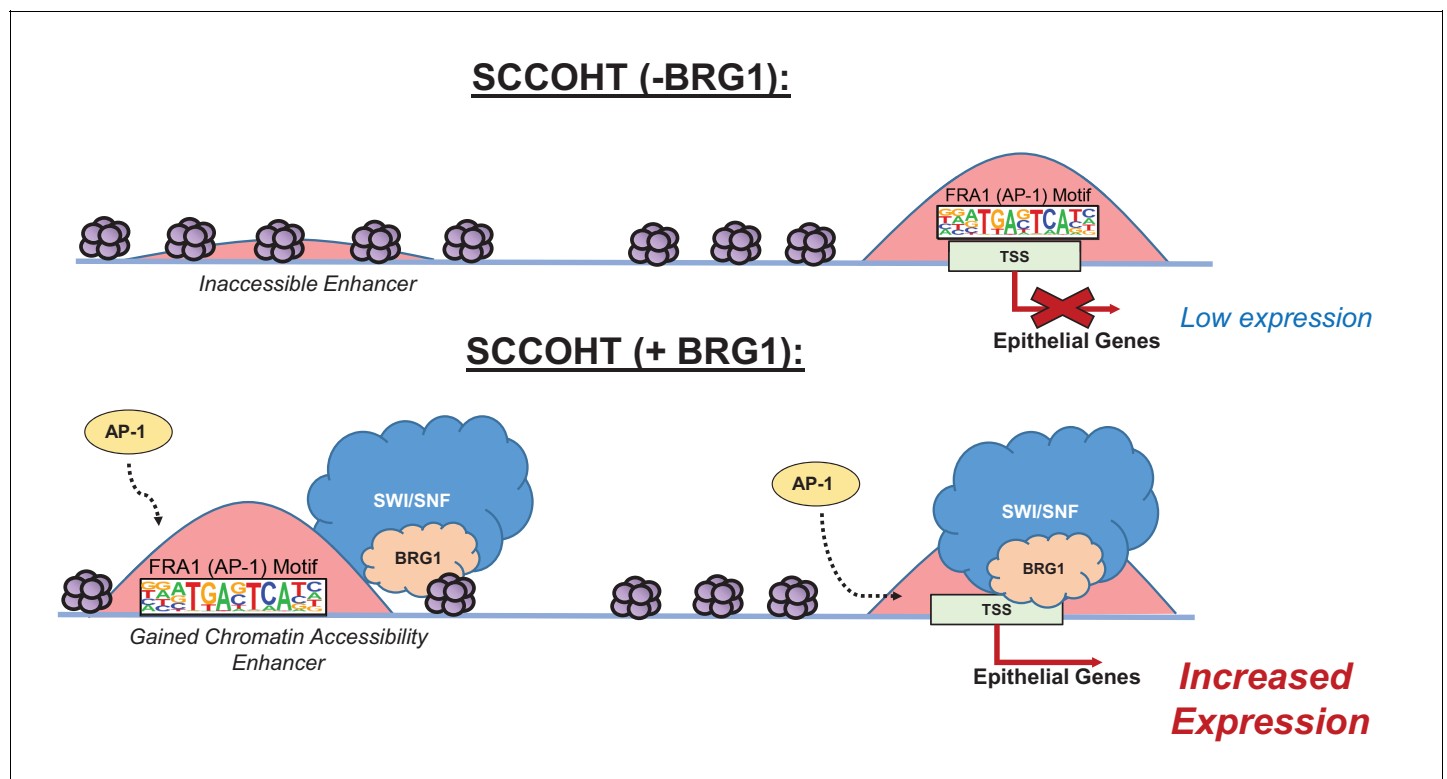

**Figure 7.** Model of BRG1 and AP-1 driving epithelial cell signature in SCCOHT tumorigenesis. Following BRG1 reexpression in SCCOHT cells, BRG1 is recruited to and opens distal chromatin enriched in AP-1 motifs. BRG1 is also recruited to promoters of epithelial genes that have AP-1 motifs. These epithelial genes are upregulated in the presence of both BRG1 and AP-1 at the gene and protein level.

## Discussion

Associations between AP-1 and various SWI/SNF complex members at enhancer regions have been previously described, such as ARID1A/B (*Mathur et al., 2017*; *Raab et al., 2015*; *Kelso et al., 2017*) in colorectal cancer and ovarian clear cell carcinoma and SNF5 (*Alver et al., 2017*) in malignant rhabdoid tumors. Specifically, BRG1 and AP-1 associations have been shown globally at enhancer sites in mouse embryonic fibroblasts (*Vierbuchen et al., 2017*), mouse mammary epithelial cells (*Hu et al., 2011*), and described specifically for CCND1 in NSCLC (*Xue et al., 2019a*) and in AML (*Michel et al., 2018*). Our results are consistent with these reports and provide the first insight into the biological significance of this mechanism showing that BRG1 reexpression in SCCOHT cell lines drives an epithelial-like gene and protein signature.

In a study by *Vierbuchen et al., 2017*, AP-1 was described as a requirement at enhancer sites to recruit the SWI/SNF complex in mouse embryonic fibroblasts (*Vierbuchen et al., 2017*). The model from Vierbuchen et al. suggests that AP-1, in cooperation with other cell-type-specific transcription factors, binds at inaccessible chromatin in enhancer regions and recruits in the SWI/SNF complex for chromatin remodeling (*Vierbuchen et al., 2017*). Much like the Vierbuchen et al. model, AP-1 has been suggested by others as a pioneer transcription factor for its ability to bind to closed chromatin and promote chromatin accessibility for the glucocorticoid receptor in mouse mammary epithelial cells (*Biddie et al., 2011*). Our model is consistent with Vierbuchen et al. where AP-1 motifs are found at presumptive enhancer sites that gain chromatin accessibility and recruit BRG1. However, our model does not delineate if AP-1 is present before BRG1 at inaccessible chromatin or if they collaboratively bind to promote chromatin accessibility at enhancer sites. The experimental timing in our model was not designed to assess the kinetics of AP-1 and SWI/SNF binding. Additionally, we found in SCCOHT-1 cells that at c-Jun sites present prior to BRG1 expression, recruitment of BRG1 did not enhance gene expression. Only sites that newly recruited c-Jun and BRG1 upregulated gene expression (*Figure 4—figure supplement 1*). Therefore, further studies are needed to determine the importance on the order of DNA binding in the AP-1 and BRG1 mechanism for epithelial-like differentiation in SCCOHT. Additionally, because the dominant-negative A-FOS is not specific to any AP-1 family member and is not known to affect the phosphorylation of any AP-1 subunits through which its activity can also occur, the AP-1 complex members required in our model will need further exploration including phosphorylation statues (*Olive et al., 1997*).

In our study, BRG1 localized to promoters and presumptive enhancer regions following BRG1 reexpression. This result is consistent with previous reports in other cell types of BRG1 recruitment to promoter regions (*Tolstorukov et al., 2013*) and to distal enhancer regions (*Alver et al., 2017*; *Hu et al., 2011*; *Morris et al., 2014*; *Agaimy et al., 2017*). These data suggest a universal role of BRG1 recruitment to promoters and enhancers, but the mechanism driving the recruitment may differ among cell types as well as different SWI/SNF complex compositions. As suggested recently, AP-1 co-localization with other transcription factors in enhancer regions depends on cell lineage (*Vierbuchen et al., 2017*), such as AP-1 and TEAD in MEFs (*Vierbuchen et al., 2017*) and breast cancer cells (*Zanconato et al., 2015*) and AP-1 and ETS family members in oesophageal adenocarcinoma (*Britton et al., 2017*). Whether or not AP-1 additionally co-localizes with another transcription factor at enhancer sites or with ZEB1/2 in the promoters of epithelial-low genes in BIN67 cells +/- BRG1 reexpression warrants further investigation.

We also explored the role of BRG1 loss in SCCOHT development. In the control BIN67 cells, we found a strong similarity to MSC by xCell cell type enrichment analysis. This finding suggests that the loss of BRG1 and the SWI/SNF complex may result in an epithelial-mesenchymal transition (EMT) or may prevent cells during early development from developing into epithelial tissue. As seen in NSCLC, another adult cancer where BRG1 loss is a progressive event, EMT is associated with poorer prognosis and disease progression (*Tsoukalas et al., 2017*; *Mahmood et al., 2017*). Given the aggressiveness and poor prognosis for SCCOHT tumors, our results suggest a more universal role of BRG1 loss in cancer in driving EMT and leading to tumor development or progression.

SCCOHT tumors are poorly differentiated and the cell of origin remains unknown. In addition to the novel biological mechanism between AP-1 and BRG1 driving epithelial-like differentiation, our work indicates that SCCOHT tumors may arise from an epithelial-like cell. We used claudin-4 as an epithelial marker because it was defined in the xCell epithelial gene list, but also because of its recent use as an epithelial IHC marker in SWI/SNF-deficient carcinoma (*Schaefer et al., 2017*). In the

initial study by *Schaefer et al., 2017*, four primary SCCOHT tumors stained for CLDN4 were negative by IHC (*Schaefer et al., 2017*). However, a second study by *Tessier-Cloutier et al., 2018*, looking at dedifferentiated and undifferentiated endometrial carcinomas, found frequent lack of CLDN4 expression despite having a known epithelial origin, suggesting that CLDN4 may not be used to infer epithelial origin in undifferentiated endometrial tumors (*Tessier-Cloutier et al., 2018*). Thus, the loss of CLDN4 in primary SCCOHT tumors does not rule out an epithelial origin. Our data is consistent with Tessier-Cloutier et al. and Schaefer et al. in that BIN67 parental cell line does not express CLDN4 at both transcript and protein levels, much like the primary SCCOHT tumors. When BRG1 is reexpressed in BIN67 cells, the cells become more differentiated and express CLDN4, suggesting an epithelial cell origin.

Recently, *Pan et al., 2019* showed that comparing SCCOHT primary tumors to normal ovary resulted in genes that were preferentially upregulated or downregulated relative to SCCOHT tumors. They used this gene set to query differentially expressed genes in BIN67 cell line +/- BRG1 or BRM reexpression (*Pan et al., 2019*). However, with the cell of origin remaining unknown for SCCOHT, normal ovary may not be the ideal tissue to compare with primary SCCOHT tumors. Of interest, some pathological analyses of primary SCCOHT samples suggest that it arises from an immature teratoma (*Kupryjańczyk et al., 2013*; *McCluggage et al., 2017*). With the addition of our data, it would suggest that an epithelial cell in one of the tissues compromising immature teratoma gives rise to a SCCOHT tumor, although we cannot define the tissue type within the immature teratoma. Therefore, studies utilizing normal ovary as the reference to compare SCCOHT primary tumors and/or cell line data +/- BRG1 or BRM may not accurately reflect the pathology of SCCOHT tumors.

In summary, we have shown that reexpression of BRG1 in the SCCOHT cell line, BIN67, induces an epithelial-like gene and protein expression. This epithelial-like signature is driven by AP-1 dependent mechanisms that require both the activity of AP-1 and BRG1 expression. Our work provides the first described biological relevance of an AP-1 and BRG1 association, specifically by driving an epithelial signature. These results yield further insights into the potential cell of origin for SCCOHT and provide a biological mechanism for the consequence of BRG1 loss in SCCOHT development.

# Materials and methods

## Key resources table

| Reagent type (species) or resource | Designation | Source or reference | Identifiers | Additional information |
|---|---|---|---|---|
| Gene (*Homo sapiens*) | BRG1 | Ensembl | ENSG00000127616 | SMARCA4 |
| Gene (*Homo sapiens*) | BRM | Ensembl | ENSG00000080503 | SMARCA2 |
| Gene (*Homo sapiens*) | JUN | Ensembl | ENSG00000177606 | JUN |
| Cell line (*Homo sapiens*) | BIN67 | Gamwell et al. (2013) doi: 10.1186/1750-1172-8-33 | SCCOHT Cell Line | Female |
| Cell line (*Homo sapiens*) | SCCOHT-1 | Otte et al. (2014) doi: 10.1186/s13023-014-0126-4 | SCCOHT Cell Line | Female |
| Cell line (*Homo sapiens*) | COV434 | *Karnezis et al., 2020*. doi: 10.1016/j.ygyno.2020.12.004 | SCCOHT Cell Line | Female |
| Cell Line (*Homo sapiens*) | A427 | ATCC HTB-53 | Non-Small Cell Lung Cancer Cell Line | Male |
| Transfected construct (*Homo sapiens*) | pBabe-puro | https://www.addgene.org/1764/ | RRID:Addgene_1764 | |

*Continued on next page*

*Continued*

| Reagent type (species) or resource | Designation | Source or reference | Identifiers | Additional information |
|---|---|---|---|---|
| Transfected construct (*Homo sapiens*) | pBJ5-BRG1 | https://www.addgene.org/17873/ | RRID:Addgene_17873 | |
| Transfected construct (*Homo sapiens*) | pCDNA3(+)-neo | Invitrogen | | |
| Lentiviral Construct (*Homo sapiens*) | pINDUCER20-FLAG-A-FOS | Ibrahim & Abudu et al. (2018). doi: 10.18632/oncotarget.26047 | | |
| Lentiviral Construct (*Homo sapiens*) | pINDUCER20-BRG1 | pINDUCER20 Backbone: https://www.addgene.org/44012/ | | |
| Antibody | Anti-Brg1 (rabbit monoclonal) | Abcam | Ab110641, RRID:AB_10861578 | CUT and RUN (1:25), WB (1:10,000) |
| Antibody | Anti-Jun (rabbit monoclonal) | Cell Signaling Technologies | 9165, RRID:AB_2130165 | CUT and RUN (1:25) |
| Antibody | Anti-Jun (rabbit ) | ThermoFisher | MA5-15172, RRID:AB_10979794 | CUT and RUN (1:25) |
| Antibody | Anti-T7 (rabbit monoclonal) | Cell Signaling Technologies | 13246, RRID:AB_2798161 | CUT and RUN (1:25) |
| Antibody | Anti-Rabbit IgG (Rabbit Monoclonal) | Cell Signaling Technologies | 3900, RRID:AB_1550038 | CUT and RUN (1:25) |
| Antibody | Anti-CD44 (Mouse Monoclonal) | Larry S. Sherman Lab and ATCC Hermes-3 | | WB (1:200); IF (1:50) |
| Antibody | Anti-Claudin-4 (Mouse Monoclonal Clone 3E2C1) | ThermoFisher (Invitrogen) | 32–9400, RRID:AB_2533096 | WB (1:1000); IHC (1:50) |
| Antibody | Anti-Ku 70/80 (Rabbit Polyclonal) | Dale Ramsden Lab - UNC Chapel Hill | | WB (1:10,000) |
| Antibody | Anti-Actin (Rabbit Monoclonal) | Sigma | A2066 | WB (1:1000) |
| Antibody | anti-FLAG (Rabbit Monoclonal) | Cell Signaling Technologies | 2368S RRID:AB_2217020 | WB (1:1000) |
| Commercial assay or kit | Fugene 6 | Promega | E269A | |
| Commercial assay or kit | Kapa Hyperprep | Roche | KK8504 | |
| Commercial assay or kit | CUT and RUN reagents | Cell Signaling Technologies | 40366 | |
| Software, algorithm | Deeptools v3.2.0 | 10.1093/nar/gkw257 | RRID:SCR_016366 | |
| Software, algorithm | Bowtie2 v 2.3.4.1 | 10.1038/nmeth.1923 | RRID:SCR_005476 | |
| Software, algorithm | Samtools v 1.9 | 10.1093/bioinformatics/btp352 | RRID:SCR_002105 | |

*Continued on next page*

*Continued*

| Reagent type (species) or resource | Designation | Source or reference | Identifiers | Additional information |
|---|---|---|---|---|
| Software, algorithm | Macs2 2.1.2 | 10.1186/gb-2008-9-9-r137 | RRID:SCR_013291 | |
| Software, algorithm | STAR (v 2.5.3a) | 10.1093/bioinformatics/bts635 | RRID:SCR_015899 | |
| Software, algorithm | FASTQC (v 0.11.5) | *Andrews, 2010*: http://www.bioinformatics.babraham.ac.uk/projects/fastqc/ | RRID:SCR_014583 | |
| Software, algorithm | DESeq2 (v1.22.2) | 10.1186/s13059-014-0550-8 | RRID:SCR_015687 | |
| Software, algorithm | HOMER (v4.10) | 10.1016/j.molcel.2010.05.004 | RRID:SCR_010881 | |
| Software, algorithm | Picard (v2.10.3 and v 2.20) | http://broadinstitute.github.io/picard/ | RRID:SCR_006525 | |
| Software, algorithm | Bedtools (v2.26.0) | 10.1093/bioinformatics/btq033 | RRID:SCR_006646 | |
| Software, algorithm | tximport (v1.14.0) | 10.18129/B9.bioc.tximport | RRID:SCR_016752 | |
| Software, algorithm | salmon(v0.8.2) | 10.1038/nmeth.4197; https://combine-lab.github.io/salmon/ | RRID:SCR_017036 | |

## Cell culture

BIN67, SCCOHT-1, COV434, A427 (ATCC), MCF-7 (ATCC), HeLa (D98OR; HeLa Subclone), 293FT (Michael Emanuale Lab UNC), PANC-1 (ATCC), and NHF+Tert (William Kaufmann Lab, UNC) cells were cultured in RPMI-1640 (Gibco #11875–093) supplemented with 10% FBS and maintained at 37° C in a humidified incubator with 5% $CO_2$. Cells were tested for mycoplasma and STR profiled for verification of identity.

## Transfections

Four biological replicates of BIN67 cells were co-transfected (FuGENE 6, Promega #E269A) with either control (pCMV-neo or pCDNA3-neo) or BRG1 (PBJ5-BRG1) plasmids at a 10:1 ratio to a plasmid containing puromycin resistance (pBABE-puro). Twenty-four hours after transfection, cells were placed on 2 µg/ml puromycin for 3 days and then harvested for analyses. Samples from each biological replicate were divided equally and processed for ATAC-seq and RNA-seq as outlined below (*Figure 1A*). Separate experiments consisting of biological triplicates were prepared for proteomic analysis and biological duplicates were prepared CUT and RUN analysis (*Figure 1A*).

SCCOHT-1 cells were transfected as described above with the modifications: cells were transfected at a 6.25:1 ratio of plasmid:pBABE-puro and selected in 0.5 µg/ml puromycin. SCCOHT-1 cells +/- BRG1 were processed for RNA-seq and CUT and RUN. COV434 cells were transfected as described above with the following modifications: cells were transfected at a ratio 10–6.25:1 of plasmid:pBABE-puro and selected in 2–3 µg/ml puromycin. COV434 cells +/- BRG1 were processed for RNA-seq.

## Inducible cell line generation

A lentivirus construct containing pINDUCER20-Flag-A-FOS was generated as previously described (*Ibrahim et al., 2018*). BIN67 pIND20-A-FOS cell lines were generated by lentiviral infection as previously described (*Wei et al., 2014*) and subsequent selection in G418 containing medium [10% TET-FREE FBS (GeminiBio #100–106) + 800 µg/ml G418 (Gibco #10131035), RPMI-1640] to create mass

populations. For all A-FOS inductions, BIN67 pIND20-A-FOS cells were induced with 1 µg/µl of doxycycline (DOX) daily with media change for 48 hr prior to transfections.

A427 pINDUCER20-BRG1 (A427 pIND20-BRG1) cell lines were generated by lentiviral infection of the pINDUCER20-BRG1 construct (Gateway Cloning) and subsequently selected in G418 containing medium +selecting in 400 µg/ml G418 to create a mass population. Following selection, the mass population was single cell cloned and screened for BRG1 expression after DOX (1 µg/µl) induction for 48 hr by western blotting. Clones that were able to induce BRG1 were pooled to create a mass pool of A427 pIND20-BRG1. For all RNA-seq studies, A427 pIND20-BRG1 cells were induced with DOX (1 µg/µl) for 48 hr.

## RNA-seq sample preparation

For studies with BIN67, SCCOHT-1, and COV434 cells, RNA from frozen cell pellets was extracted using the Zymo Quick-RNA Miniprep Kit (Zymo cat# R1054). Total RNA was prepared using the Illumina TruSeq mRNA v2 kit with an input of 500 ng of RNA for each sample to produce unstranded RNA libraries following the manufacturer's protocol. Final RNA libraries were quantified using the Qubit High Sense Reagent kit and Agilent Tapestation HSD1000 tapes. Libraries were equimolar pooled and paired end sequenced across three lanes of a HiSeq4000 (paired end x 78 bp).

For RNA-seq analysis of the BIN67 pIND20-A-FOS and A427 pIND20-BRG1 cell lines, RNA was extracted from frozen cell pellets in biological duplicate (BIN67 pIND20-A-FOS) and biological triplicate (A427 pIND20-BRG1) using Zymo Quick-RNA Miniprep Kit. Library preparations and RNA sequencing was performed by Novogene Inc.

## RNA-seq data analysis

For each sample, fastq files were initially analyzed by fastqc (v0.11.5). Reads were aligned with STAR (v2.5.3a) to GENCODE v24 using the two-pass method. Aligned reads were quantified by salmon (v0.8.2) for transcripts per million (TPM) counts. TPMs were summarized to gene level using tximport (v1.10.1) in R. Differentially expressed genes between BRG1 and Control samples using DESeq2 (v1.22.2). Counts summarized to gene level from RNA-seq results were VST normalized using DESeq2. VST normalized counts were provided as input for gene set enrichment analysis performed by GSEA (v3.0) for both the Hallmark and GO (c5 all) gene sets from MSigDB (v6.0). Cell type enrichment analysis was performed by xCell (*Aran et al., 2017*) using gene and library scaled TPMs (tximport, countsFromAbundance = lengthScaledTPMs) as input. xCell heatmap was generated using the final score file provided by xCell analysis. Transcription factor motif analysis at the promoter of BIN67 xCell genes was performed with HOMER(v4.10) findMotifs.pl command with default settings. IGV tracks (bigwig) were generated on the merged biological replicates by deeptools (v3.2.) bamCoverage with a binsize of 30 and normalized using RPKMs.

Epithelial signature scoring was performed by the R package, singscore, for the curated epithelial signature (n = 171 unique genes) and using gene and library scaled TPMs (tximport, countsFromAbundance = lengthScaledTPMs) as input.

## ATAC-seq sample preparation

Four biological replicates were processed for ATAC-seq following standard protocol (*Buenrostro et al., 2013*) with modifications: increasing the cell number per sample to 100,000 cells, the use of Zymo ChiP DNA Clean and Concentrator Kit (Zymo Cat #D5205) in place of Qiagen MinElute, and lastly size selected with AMPure XP Beads (1.8x volume; Agencourt #A63880). For each biological replicate, samples were done in technical triplicate and run on 6% TBE gel to assay for nucleosome banding. The technical sample with the most pronounced nucleosome banding within each biological sample was selected for future studies and were then analyzed by Agilent Tapestation (UNC Vironomics Core) for molarity content. Samples were equimolar pooled (5 nM each) and paired end sequenced across two lanes on a Illumina HiSeq2500 (Paired end x 50 bp; UNC High Throughput Sequencing Facility).

## ATAC-seq data analysis

For each sample, fastq files from both sequencing lanes were merged with cat command. Fastq files were initially analyzed by fastqc (v0.11.5). Adaptor sequences were trimmed from reads using

trimgalore (v0.6.2) with –nextera argument. Reads were aligned with Bowtie2 (v2.3.2) to UCSC hg38 using –x 2000. Aligned mitochondrial reads were eliminated. Aligned reads were sorted with samtools (v1.9) and duplicates were removed using picard (v2.10.3). Subsequently, hg38 blacklisted regions (https://sites.google.com/site/anshulkundaje/projects/blacklists) (ENCODE Project Consortium, 2012) and BRG1 plasmid reads were removed using bedtools (v2.26.0) intersect. Biological replicates for each condition were merged using samtools merge (v1.9). Peaks were called on the merged replicates using MACS (v2.1.0) without model (–nomodel). Called peaks from both control and BRG1 conditions were merged into a union peak set using bedtools merge (v.2.26.0). Reads under the union peak set in each biological sample were counted using featureCounts (subread v1.6.3). Differential peak analysis was performed using DESeq2 (v1.22.2). Peak annotation was performed with HOMER (v4.10). Motif analysis on gained peaks was performed with HOMER (v4.10) with static peaks as background input. For static and lost peaks, motif analysis was performed with HOMER (v4.10) using the default for background. IGV tracks (bigwig) for the merged biological replicates were generated by deeptools (v3.2.) bamCoverage with a binsize of 30 and normalized using RPKMs. FRA1 motifs in the genome were annotated with HOMER (v4.10) scanMotifGenomeWide.pl tool.

## The Cancer Genome Atlas (TCGA) data integration: ATAC-seq, xCell results, BRG1 mutation frequency

Cancer type-specific ATAC-seq peak sets for TCGA tumors were downloaded from supplemental data published in (https://gdc.cancer.gov/about-data/publications/ATACseq-AWG) (Corces et al., 2018). The overlap of ATAC-seq peaks gained by BRG1 reexpression in BIN67 cells (Gained ATAC-seq Peaks; n = 4023 peaks) with TCGA ATAC-seq peaks was determined by bedtools intersect. The percentage of gained ATAC-seq peaks that overlapped with a TCGA ATAC-seq peak was determined for each tumor type (percentage overlap = [# of gained peaks that overlap per TCGA tumor type/4023] x 100).

Calculated xCell scores for TCGA tumors were downloaded from supplemental data published in Aran et al., 2017 and averaged per TCGA tumor type.

Data from TCGA was downloaded from the Genomic Data Commons using TCGABiolinks to identify mutation status of BRG1 (SMARCA4). Silent mutations of SMARCA4 were eliminated for this study. The fraction of tumors with BRG1 mutations was calculated as the total mutated tumors divided by the total number of tumors available and reported as a percentage.

## CUT-and-RUN sample preparation

CUT and RUN was performed as previously described (Skene and Henikoff, 2017) with minor modifications (Protocol available through protocols.io; dx.doi.org/10.17504/protocols.io.zcpf2vn). Following transfection of BRG1 plasmids into BIN67 or SCCOHT-1 cells, as described above, 50,000–200,000 were harvested using accutase and bound to Concanavalin A coated beads (Epicypher). Cells bound to beads were then incubated with antibody overnight at 4°C (1:25 dilution of BRG1 (Abcam BRG1 #ab110641, EPNCIR111A)), 1:25 dilution of JUN (Cell Signaling #9165 or ThermoFisher #MA5-15172), or 1:25 for T7 Rabbit Monoclonal (Cell Signaling #13246, used with SCCOHT-1) or Rabbit Monoclonal IgG as control (Cell Signaling #3900, used with BIN67). Following overnight incubation with antibody, cells were washed four times in Digitonin Wash buffer, resuspended in Digitonin Wash buffer containing protein A/G MNase (Cell Signaling Technologies) and then incubated for 1 hr at 4°C. After binding of protein A/G-MNase, cells were again washed four times before resuspension in ice cold Digitonin Wash Buffer. Calcium chloride was added to final concentration of 2 mM to induce digestion of chromatin and the reaction was allowed to proceed at 0°C for 30 min before being stopped by the addition of STOP buffer to a final concentration of (170 mM NaCl, 10 mM EDTA, 2 mM EGTA, 0.05% Digitonin, 50 µg/mL RNase). DNA was extracted by incubating cells at 37°C for 30 min and purified using a Zymo Clean and Concentrator ChIP Kit followed by library prep (Kapa HyperPrep). Libraries were sequenced on an Illumina Nextseq 500 (Paired-end 37 base-pair reads).

## CUT-and-RUN analysis

Reads were trimmed using trim_galore (version 0.6.2), and aligned to hg38 using bowtie2 (version 2.3.4.1) using –very-sensitive-local –no-mixed –no-discordant –no-unal –dovetail. Reads were indexed and sorted using samtools (v1.9) and duplicates were removed using picard (v 2.20). Genome coverage was calculated using Deeptools bamCoverage (v3.2.0) using –binSize 30 – smoothLength 60 –extendReads –normalizeUsingRPGC –ignoreDuplicates. These tracks were further processed by calculating the ratio between IP and IgG and then averaged using wiggletools and wigToBigWig. Peaks were called using Macs2 (v2.12) for each antibody relative to IgG as control. We annotated BRG1 and c-Jun peaks to their nearest genes using ChIPpeakAnno (3.22.4) to assess changes in gene expression associated with BRG1 and/or c-Jun occupancy. All heatmaps were drawn using Deeptools (v3.2.0).

## Proteomic analysis

Total protein preparation from frozen cell pellets ($2 \times 10^6$ cells per sample) was performed as previously described (*Kovalchik et al., 2019*). Tryptic peptides from each sample were individually labeled with TMT 11-plex labels, pooled, and fractionated by high pH RP-HPLC into 48 fractions and concatenated into 12 fractions, desalted, orthogonally separated, and analyzed using an Easy-nLC 1000 coupled to a Thermo Scientific Orbitrap Fusion mass spectrometer operating in MS3 mode.

MS/MS were searched against the UniProt reference proteome (20,247 sequences, 2018/08/03) using Sequest HT algorithm through the Proteome Discoverer suite (v2.1.1.21, Thermo Scientific). Precursor and fragment ion tolerance were set to 20 ppm and 0.8 Da, respectively. Dynamic modifications included Oxidation (+15.995 Da, M), Acetylation (+42.011 Da, N-Term), and static modification included Carbamidomethyl (+57.021 Da, C) and TMT (+229.163 Da, K, N-Term). Peptide-to-spectrum matches (PSMs) were calculated using Percolator by searching the results against a decoy sequence set; only PSMs with FDR < 1% were retained in the analysis.

Downstream data processing was performed in R software, with PSMs assigned to a unique protein ID and not part of a common contaminant list were median aggregated into unique peptides. In order to detect which proteins were differentially expressed between conditions, PECA (*Suomi et al., 2015*) analysis was performed at the peptide level, using median normalization and modified t-statistic parameters. Only proteins with two or more unique peptides were considered for further analysis. GSEA on the log2 peptide counts was performed by GSEA (v3.0) for both the Hallmark and GO (c5 all) gene sets from MSigDB (v6.0).

## Protein extractions

Cells were washed in the dish with cold 1X PBS on ice. Cells were lysed and harvested using a cell scraper in RIPA High-Salt Extraction Buffer (20 mM Tris pH 7.5, 400 mM NaCl, 0.5% NP-40, and 0.1% SDS) supplemented with 1X protease inhibitors (Roche cat# 11836170001). Lysates were incubated on ice for 30 min and then pelleted at 14,000 RPMs for 30 min. Supernatant containing extracted protein was removed and saved. Protein was quantified using the Pierce BCA Protein Assay Kit (Thermo-Scientific cat# 23227).

## Western blotting

Extracted protein (7 μg) was separated on a NuPAGE 4–12% Bis-Tris Gel (Invitrogen #NP0321BOX) in MOPS SDS Running Buffer (Nupage #NP0001) at 184V for 45 min. Proteins were transferred to a nitrocellulose membrane (Thermo-Scientific #88018) using 2x Bis-Tris Transfer Buffer (Boston Bio-Products # BP-193) with 15% Ethanol at 24V for 1 hr. Transfer was confirmed with ponceau staining and blocked in 5% Non-Fat Dry Milk (NFDM) in 1x PBS for 1 hr. The membrane was washed using 1x PBS ($3 \times 10$ min), cut, and incubated overnight at 4˚C in the following primary antibodies: BRG1 (1:10,000 - Abcam #ab110641), Ku 70/80 (1:10,000 - a gift from Dale Ramsden Lab, UNC), FLAG (1:1,000 - CST #2368S), CD44 (1:200 - Hermes 3) (*Strobeck et al., 2001*), and Claudin-4 (1:1,000 - Invitrogen #32–940). Following primary incubation, membranes were washed in 1X PBS ($3 \times 10$ min) and incubated in LiCOR Secondary (IR Dye 800cw; Goat anti-Rabbit: cat# 926–32211 or Donkey anti-Mouse: cat# 926–32212) diluted into 5% NFDM 1X PBS (1:5000 to 1:10,000) for 1 hr at room temperature. Blots were washed with 1x PBS ($3 \times 10$ min) and imaged on LiCOR Odyssey machine at High Quality and a resolution of 169 μm.

## Tissue microarray (TMA) construction and immunohistochemistry

The epithelial ovarian cancer TMA was built using cases from the archives of Vancouver General Hospital. Duplicate 0.6 or 1.0 mm cores from each case were used for TMA construction, as described previously (*Alkushi et al., 2007*). The SCCOHT TMA was built as previously described (*Karnezis et al., 2016*). TMAs were cut at 4 µm thickness onto Superfrost+ glass slides, and were processed using the Ventana Discovery Ultra machine (Ventana Medical Systems, Tucson, AZ, USA) as per manufacturer's protocol with proprietary reagents.

Primary Immunostaining includes CLDN4 (Zymed, clone 3E2C1, 1:50). The CLDN4 staining intensity and percentage of positive cells was scored by a pathologist (BTC). The intensity was defined as negative (<1% of tumor cells showing definite, membrane staining), weak (1), moderate (2), or strong (3). The H-score was calculated using intensity x percentage.

For CD44 immunofluorescence, paraffin sections were deparaffinized in xylene and then rehydrated in graded alcohols. Slides were heated in sodium citrate buffer (10 mM, pH 6.0) for 5 min in a microwave for antigen retrieval. Sections were blocked with 5% goat normal serum, then incubated with primary antibody against CD44 (1:50 Hermes-3; mouse monoclonal from ATCC) overnight at 4° C. The next day, sections were incubated with the fluorochrome-conjugated goat anti-mouse IgG (Alexa 488, 1:1000, Molecular Probes Inc) and counterstained with Hoechst 33342 (1:10,000, Molecular Probes) to label cell nuclei. Stained sections were mounted and examined by fluorescence microscopy using an Olympus VS120 SlideScanner using a 2x/0.08NA Olympus Plan ApoM objective for overview scans, and an 20x/0.75NA Olympus PlanSApo objective for detailed fluorescent scans. Images were acquired as 10 µM Z-stacks with a 0.5 µM interval (20 slices total). Z projection image processing and cell counts were performed using ImageJ software (National Institutes of Health). The number of total nuclei and CD44-positive cells in each sample were counted and the percentage of CD44-positive cells in each sample was calculated.

## Acknowledgements

The authors thank the late Dr. Michele Fluck for guidance with the pIND20-FLAG-A-FOS plasmid, Dr. Barbara Vanderhyden (University of Ottawa) for providing the BIN67 cell line, Dr, Ralf Hass (Medical University of Hannover) for providing the SCCOHT-1 cell line, Dr. Mikko Anttonen (University of Helsinki) for providing the COV434 cell line, UNC High Throughput Sequencing Facility and the Sequencing and Genomic Technologies Shared Resource at Duke University. This work was supported by research funds from NIH R01CA195670 to DGH, JMT, and BEW, NIH P30 CA016086 (LCCC Cancer Center Support Grant), DOD W81XWH-19-1-0423 to JRR, National Institutes of Environmental and Health Sciences T32ES007126 to VN, and by the Integrated Pathology Core at the Oregon National Primate Research Center (ONPRC) which is supported by NIH Award P51 OD 011092.

## Additional information

### Funding

| Funder | Grant reference number | Author |
|---|---|---|
| National Institutes of Health | R01CA195670 | David G Huntsman<br>Jeffrey M Trent<br>Bernard E Weissman |
| National Institutes of Health | P30CA016086 | Joel S Parker |
| National Institutes of Health | T32ES007126 | Vinh Nguyen |
| Department of Defense | W81XWH-19-1-0423 | Jesse R Raab |
| National Institutes of Health | P51 OD 011092 | Larry S Sherman |

The funders had no role in study design, data collection and interpretation, or the decision to submit the work for publication.

## Author contributions
Krystal Ann Orlando, Conceptualization, Data curation, Software, Formal analysis, Validation, Investigation, Visualization, Methodology, Writing - original draft, Writing - review and editing; Amber K Douglas, Data curation, Formal analysis, Validation, Investigation, Visualization, Methodology, Writing - review and editing; Aierken Abudu, Resources, Writing - review and editing; Yemin Wang, Resources, Data curation, Project administration, Writing - review and editing; Basile Tessier-Cloutier, Investigation, Methodology, Writing - review and editing; Weiping Su, Formal analysis, Investigation, Methodology, Writing - review and editing; Alec Peters, Rayvon Moore, Investigation, Methodology; Larry S Sherman, Formal analysis, Supervision, Investigation, Methodology, Writing - review and editing; Vinh Nguyen, Formal analysis, Investigation, Methodology; Gian Luca Negri, Shane Colborne, Resources, Data curation, Software, Formal analysis, Investigation, Visualization, Methodology, Writing - review and editing; Gregg B Morin, Resources, Data curation, Software, Supervision, Visualization, Methodology, Project administration, Writing - review and editing; Friedrich Kommoss, Resources; Jessica D Lang, Validation, Investigation, Project administration, Writing - review and editing; William PD Hendricks, Conceptualization, Funding acquisition, Project administration, Writing - review and editing; Elizabeth A Raupach, Writing - review and editing; Patrick Pirrotte, Visualization, Writing - review and editing; David G Huntsman, Bernard E Weissman, Conceptualization, Resources, Supervision, Funding acquisition, Methodology, Project administration, Writing - review and editing; Jeffrey M Trent, Conceptualization, Resources, Supervision, Funding acquisition, Project administration, Writing - review and editing; Joel S Parker, Conceptualization, Software, Supervision, Visualization, Methodology, Project administration, Writing - review and editing; Jesse R Raab, Conceptualization, Resources, Data curation, Software, Formal analysis, Supervision, Validation, Investigation, Visualization, Methodology, Project administration, Writing - review and editing

## Author ORCIDs
Krystal Ann Orlando ![ORCID] https://orcid.org/0000-0001-5790-8843
Gian Luca Negri ![ORCID] http://orcid.org/0000-0001-7722-8888
Jessica D Lang ![ORCID] http://orcid.org/0000-0001-9700-4785
Jesse R Raab ![ORCID] https://orcid.org/0000-0001-6387-8994
Bernard E Weissman ![ORCID] https://orcid.org/0000-0002-1827-2309

## Decision letter and Author response
Decision letter https://doi.org/10.7554/eLife.59073.sa1
Author response https://doi.org/10.7554/eLife.59073.sa2

---

# Additional files

## Supplementary files
• Supplementary file 1. RNA-seq and Proteomics differential expression results for BIN67 +/- BRG1 reexpression. A, Table of DESeq2 results for RNA-seq BIN67 +/- BRG1 samples. Log2FoldChange = BIN67/Control. B, Table of PECA analysis results for proteomics BIN67 +/- BRG1.

• Supplementary file 2. Transcription factor motif results for ATAC-seq gained peaks. Table of transcription factor motif analysis results for ATAC-seq gained peaks described in *Figure 3e*

• Supplementary file 3. RNA-seq differential expression results for BIN67 +/- BRG1 +/- A FOS. A, Table of DESeq2 results for BIN67 pIND20-FLAG-A-FOS, -DOX Conditions (absent A-FOS), +/- BRG1used in volcano plot *Figure 6*. Log2Foldchange = BRG1/Control. B, Table of DESeq2 results for BIN67 pIND20-FLAG-A-FOS, Control transfected, +/- DOX (A-FOS) used in volcano plot *Figure 6*. Log2Foldchange = +DOX/-DOX. B, Table of DESeq2 results for BIN67 pIND20-FLAG-A-FOS, BRG1 transfected, +/- DOX (A-FOS) used in volcano plot *Figure 6*. Log2Foldchange = +DOX/-DOX.

• Supplementary file 4. RNA-seq differential expression results for SCCOHT-1 and COV434 +/- BRG1 reexpression. A. Table of DESeq2 Results for SCCOHT-1 cells +/- BRG1. B. Table of DESeq2 results for COV434 +/- BRG1

• Supplementary file 5. ATAC sites used in analysis of BRG1 and c-Jun localization. A. ATAC sites gained following expression of BRG1. B. ATAC sites that overlap a Fra1 motif, used to identify protein localization relative to motif location.

• Supplementary file 6. Peaks identified in CUT and RUN analysis. Table of output from macs2 peak calling on each CUT-and-RUN experiment for BRG1 and c-Jun in BIN67 and SCCOHT-1 cells.

• Supplementary file 7. Transcription factor motif results for BRG1 peaks found in BIN67 and SCCOHT-1. Motif analysis results from homer to identify known transcription factor motifs enriched at BRG1 peak locations.

• Transparent reporting form

### Data availability

Raw fastq files and processed data have been deposited in Gene Expression Omnibus (GEO) database with the accession number: GSE151026. Proteomics data was deposited in PRIDE database (accession #PXD014134).

The following datasets were generated:

| Author(s) | Year | Dataset title | Dataset URL | Database and Identifier |
|---|---|---|---|---|
| Orlando KA, Douglas AK, Abudu A, Wang Y, Tessier-Cloutier B, Su W, Peters A, Sherman LS, Moore R, Nguyen V, Negri GL, Colborne S, Morin GB, Kommoss F, Lang JD, Hendricks WP, Raupach EA, Pirrotte P, Huntsman DG, Trent JM, Parker JS, Raab JR, Weissman BE | 2020 | Re-expression of SMARCA4/BRG1 in Small Cell Carcinoma of Ovary, Hypercalcemic Type (SCCOHT) promotes an epithelial-like gene signature through an AP-1-dependent mechanism | https://www.ncbi.nlm.nih.gov/geo/query/acc.cgi?acc=GSE151026 | NCBI Gene Expression Omnibus, GSE151026 |
| Orlando KA, Douglas AK, Abudu A, Wang Y, Tessier-Cloutier B, Su W, Peters A, Sherman LS, Moore R, Nguyen V, Negri GL, Colborne S, Morin GB, Kommoss F, Lang JD, Hendricks WP, Raupach EA, Pirrotte P, Huntsman DG, Trent JM, Parker JS, Raab JR, Weissman BE | 2020 | SMARCA4 regulates an epithelial-like gene signature through AP-1 driven mechanisms in Small Cell Carcinoma of Ovary-Hypercalcemic Type | https://www.ebi.ac.uk/pride/archive/projects/PXD014134 | PRIDE, PXD014134 |

The following previously published datasets were used:

| Author(s) | Year | Dataset title | Dataset URL | Database and Identifier |
|---|---|---|---|---|
| Pan J, McKenzie ZM, D'Avino AR, Mashtalir N, Lareau CA, St Pierre R, Wang L, Shilatifard A, Kadoch C | 2019 | The ATPase module of mammalian SWI/SNF family complexes mediates subcomplex identity and catalytic activity-independent genomic targeting | https://www.ncbi.nlm.nih.gov/geo/query/acc.cgi?acc=GSE117735 | NCBI Gene Expression Omnibus, GSE117735 |
| Xue Y, Johnson RM, Foulkes WD, Huang | 2019 | CDK4/6 inhibitors target SMARCA4-determined cyclin D1 | https://www.ncbi.nlm.nih.gov/geo/query/acc. | NCBI Gene Expression Omnibus, |

| | | | | |
|---|---|---|---|---|
| S | | deficiency in hypercalcemic small cell carcinoma of the ovary (I) | cgi?acc=GSE120297 | GSE120297 |
| Song S, Nguyen V, Schrank T, Mulvaney K, Walter V, Wei D, Orvis T, Desai N, Zhang J, Hayes DN, Zheng Y, Major MB, Weissman BE | 2020 | Loss of SWI/SNF Chromatin Remodeling Alters NRF2 Signaling in Non-Small Cell Lung Carcinoma | https://www.ncbi.nlm. nih.gov/geo/query/acc. cgi?acc=GSE162611 | NCBI Gene Expression Omnibus, GSE162611 |

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
