## [Decision Letter]

**Acceptance summary:**

This manuscript describes a multi-omics approach to assess the activity of BRG1 in small cell carcinoma of the ovary hypercalcemic type (SCCOHT). SCCOHT is an aggressive cancer that is characterized by SMARCA4/BRG1 mutations and a loss of protein expression. The authors restore BRG1 in SCCOHT cells and determine the transcriptional, proteomic, and chromatin consequences. The authors find that BRG1 expression promotes an AP-1-dependent epithelial like-cell differentiation, suggesting that SCCOHT potentially originates from an epithelial cell. The major strengths of the study include an investigation into a rare ovarian cancer, multi-omic approaches, novel findings, and correlation to previous and independent datasets, method transparency, and rigor.

**Decision letter after peer review:**

Thank you for submitting your article "Re-expression of SMARCA4/BRG1 in SCCOHT promotes an epithelial-like gene signature through an AP-1-dependent mechanism" for consideration by *eLife*. Your article has been reviewed by three peer reviewers, and the evaluation has been overseen by Maureen Murphy as the Senior and Reviewing Editor. The following individuals involved in review of your submission have agreed to reveal their identity: Xiaobing Shi (Reviewer #1); Benjamin Bitler (Reviewer #3).

The reviewers have discussed the reviews with one another and the Reviewing Editor has drafted this decision to help you prepare a revised submission.

Summary

Weissman and colleagues use a multi-omics approach to assess the key genes and pathways for BRG1 in small cell carcinoma of the ovary hypercalcemic type (SCCOHT). SCCOHT is an aggressive cancer that is characterized by SMARCA4/BRG1 mutations and a loss of protein expression. Currently, the SCCOHT cell of origin is unknown, and determining the cell of origin will allow for more appropriate controls and improved understandings of the disease etiology. They restore BRG1 in the BIN67 SCCOHT cell line and determine the transcriptional, proteomic, and chromatin consequence of re-expressing BRG1. The data supports that BRG1 expression promotes an AP-1-dependent epithelial like-cell differentiation, suggesting that SCCOHT potentially originates from an epithelial cell. The major strengths of the study include an investigation into a rare ovarian cancer, multi-omic approaches, novel findings, and correlation to previous and independent datasets, method transparency, and rigor.

Essential revisions

1) Given that AP-1 controls transcription of a large number of genes, the number of the genes that affected by pIND20-FLAG-A-FOS was unexpectedly small. ChIP-seq experiments using a c-fos antibody would help to test if 1) the predicted AP-1 binding motifs (shown in Figure 3E and 3F; Figure 4C) are indeed bound by c-fos in SCCOHT; 2) the genes identified by RNA-seq (shown in Figure 5D and 5E) are direct targets of c-fos. Alternatively the authors could perform targeted ChIP analyses for Fos in the BIN67 cell line, +/- BRG1, on selected promoters, such as the CD44 and claudin promoters.

2) One limitation of the study is the use of a single cell line for the majority of studies, and the use of transient transfection and overexpression of BRG1. It is requested that the authors replicate a few more of the key findings in another SCCOHT cell line, such as the SCCOHT-1 line referred to in the text. For example, can the authors replicate the key observations using dominant negative AP-1 in the additional SCCOHT-1 cell line, OR perform the ChIP described above for Fos +/- BRG1 on CD44 and claudin in the SCCOHT-1 (additional) cell line.

3) To alleviate concerns about the use of transfection, can the authors silence BRG1 and show whether this results in predicted changes in gene expression of AP-1 targets, or EMT?

4) Does overexpression of BRG1 in any cell type (e.g. PANC1) drive an epithelial-like transcriptome /proteome? This could be addressed experimentally or in the Discussion.

---

## [Author Response]

Essential revisions1) Given that AP-1 controls transcription of a large number of genes, the number of the genes that affected by pIND20-FLAG-A-FOS was unexpectedly small. ChIP-seq experiments using a c-fos antibody would help to test if 1) the predicted AP-1 binding motifs (shown in Figure 3E and 3F; Figure 4C) are indeed bound by c-fos in SCCOHT; 2) the genes identified by RNA-seq (shown in Figure 5D and 5E) are direct targets of c-fos. Alternatively the authors could perform targeted ChIP analyses for Fos in the BIN67 cell line, +/- BRG1, on selected promoters, such as the CD44 and claudin promoters.

We have now included CUT&RUN experiments for c-Jun in the original cell line (BIN67) as well as a second SCCOHT cell line (SCCOHT-1). In both cell lines, BRG1 sites induced following BRG1 re-expression recruit c-Jun (Figure 4D, E). BIN67 cells have very low levels of c-Jun binding prior to BRG1 re-expression. Thus, with our ATAC-seq data, we show that Fra1 motifs recruit both BRG1 and JUN to proximal and distal sites of genes. However, chromatin opening and histone modification changes occur mostly at distal regions (Figure 4G). Notably, there are some differences in c-Jun occupancy between cell lines. In HT1, considerable c-Jun binding exists prior to BRG1 re-expression. In contrast, BIN67 cells display little/no c-Jun occupancy prior to BRG1 re-expression (Figure 4G, and Figure 4—figure supplement 1). We consider this difference in the Discussion, and highlight the possibility that different AP-1 family members are expressed in different cell lines (shown for BIN67/HT1 in Figure 4—figure supplement 1C, and Figure 6—figure supplement 1). This finding additionally emphasizes the importance of using the dominant negative A-FOS in our experiments.

2) One limitation of the study is the use of a single cell line for the majority of studies, and the use of transient transfection and overexpression of BRG1. It is requested that the authors replicate a few more of the key findings in another SCCOHT cell line, such as the SCCOHT-1 line referred to in the text. For example, can the authors replicate the key observations using dominant negative AP-1 in the additional SCCOHT-1 cell line, OR perform the ChIP described above for Fos +/- BRG1 on CD44 and claudin in the SCCOHT-1 (additional) cell line.

We have now included CUT&RUN experiments for BRG1 and c-Jun in a second cell line (SCCOHT-1). We found that c-Jun is bound at BRG1 opened chromatin sites and that motifs associated with BRG1 occupancy are enriched for AP-1 family members, supporting our initial result in BIN67 (Figure 4). In addition, we performed RNA-seq in SCCOHT-1 and a third SCCOHT cell line, COV434, and found the epithelial gene expression program is activated by BRG1 re-expression (Figure 2).

3) To alleviate concerns about the use of transfection, can the authors silence BRG1 and show whether this results in predicted changes in gene expression of AP-1 targets, or EMT?

There are no SCCOHT cell lines expressing BRG1 that could be used in this experiment. The broader question- whether methods beside transient transfection show gene expression changes of AP-1 targets or epithelial transcription- is addressed in point 4 below.

4) Does overexpression of BRG1 in any cell type (e.g. PANC1) drive an epithelial-like transcriptome /proteome? This could be addressed experimentally or in the Discussion.

To address this, we include re-analysis of published data as well as new data to show that re-expression of BRG1 activates an epithelial signature. Data in Figure 2G show this activation for other SCCOHT cells and Figure 3A in a lung adenocarcinoma (LUAD) cell line (A427). Additionally, silencing of BRG1 but not BRM in the LUAD cell line H358 cell line causes a decrease in the epithelial signature suggesting that this may be a general across multiple cell types (Figure 3B).